# 🥐 CroissantLLM: A Truly Bilingual French-English Language Model

Manuel Faysse[1,5]    Patrick Fernandes[6,8,11]    Nuno M. Guerreiro[2,5,6,8]    António Loison[1]
Duarte M. Alves[6,8]    Caio Corro[9]    Nicolas Boizard[4,5]    João Alves[2]    Ricardo Rei[2,7,8]
Pedro H. Martins[2]    Antoni Bigata Casademunt[10]    François Yvon[9]    André F.T. Martins[2,6,8]
Gautier Viaud[1]    Céline Hudelot[5]    Pierre Colombo[3,5]

[1]Illuin Technology    [2]Unbabel    [3]Equall    [4]Diabolocom    [5]MICS, CentraleSupélec
[6]Instituto de Telecomunicações, Lisboa    [7]INESC-ID, Lisboa    [8]Instituto Superior Técnico, Universidade de Lisboa
[9]Sorbonne Université, CNRS, ISIR, Paris    [10]Imperial College London
[11]Language Technologies Institute, Carnegie Mellon University

Reviewed on OpenReview: `https://openreview.net/forum?id=uA19Xo1o31`

## Abstract

We introduce CroissantLLM, a 1.3B language model pre-trained on a set of 3T English and French tokens, to bring to the research and industrial community a high-performance, fully open-sourced bilingual model that runs swiftly on consumer-grade local hardware. To that end, we pioneer the approach of training an intrinsically bilingual model with a 1:1 English-to-French pretraining data ratio, a custom tokenizer, and bilingual finetuning datasets. We release the training dataset, notably containing a French split with manually curated, high-quality, and varied data sources. To assess performance outside of English, we craft a novel benchmark, FrenchBench, consisting of an array of classification and generation tasks, covering various orthogonal aspects of model performance in the French Language. Additionally, rooted in transparency and to foster further Large Language Model research, we release codebases, and dozens of checkpoints across various model sizes, training data distributions, and training steps, as well as fine-tuned Chat models, and strong translation models. We evaluate our model through the FMTI framework (Bommasani et al., 2023) and validate 81 % of the transparency criteria, far beyond the scores of even most open initiatives. This work enriches the NLP landscape, breaking away from previous English-centric work to strengthen our understanding of multilingualism in language models.

## 1 Introduction

Large Language Models (LLM)[1] have taken over the Natural Language Processing (NLP) landscape in the past years. Although a few proprietary models are still considered to run ahead of the pack (OpenAI et al., 2023), open weights models such as Llama (Touvron et al., 2023a;b), Qwen (Bai et al., 2023a) or Mistral (Jiang et al., 2023; 2024) are rapidly bridging the performance gap. However, widespread industrial and research adoption of such technology remains challenging for several reasons, including the lack of transparency in the data collection and training processes, the scarcity of existing resources outside of English, and the large-scale and costly nature of existing high-performing models.

**Lack of transparency.** State-of-the-art models, both proprietary and open-weights are often designed and trained by heavily investor-backed companies, that aim to retain a moat by keeping their training data mix and strategy secret, hindering the rest of the field's ability to fully study and understand these models. This lack of transparency, ranging from training set composition to lack of evaluation or unclear usage policies,

---

[1]See Appendix E for a definition.

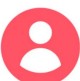
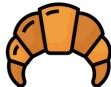

Explain the French expression "poser un lapin"?

"Poser un lapin" is an expression that means to delay or avoid something that one is expected to do, usually by not showing up or not doing it at all. It is often used in jest or sarcastically, but it can also have a more serious meaning. In French, the expression is often used to describe someone who is unreliable or unreasonable, and who is not willing to follow through on their commitments. It can also be used to describe someone who is rude or dismissive of others' needs.

Figure 1: Conversation example with CroissantLLMChat

has been characterized by previous works, such as those by Bommasani et al. (2023) and Casper et al. (2024), pushing for full transparency as a key component for safe model development and use. The dangers of closed, non-auditable datasets have been exemplified by recent findings showcasing the potential dangers of dataset contamination, whether intentional (Hubinger et al., 2024) or not.[2] Furthermore, legal questions arise surrounding data ownership in LLM training corpora (NewYorkTimes, 2023; Samuelson, 2023) and recent developments in the political landscape, regarding AI (EU AI Act, US senate hearings)[3] have further emphasized the importance of transparent approaches, both from a legal perspective and to build user trust.

**Bias towards English.** Although the exact training mix of most well-performing LLMs is not publicly available information, most large models are trained on very English-centric corpora (Touvron et al., 2023a). This is the consequence of the important amount of English resources compared to other languages, both in terms of data availability and benchmarks. As could be expected, all publicly available LLMs display a large performance disparity between English and non-English languages when evaluated on downstream tasks (Bandarkar et al., 2023). Moreover, cultural knowledge and biases are mechanically integrated through the training data, leading to models with greater knowledge of American events or biases (Bender et al., 2021; Santurkar et al., 2023). This puts non-English users at a disadvantage when it comes to language model usage and adoption. While the non-English NLP community has produced multilingual datasets (Soboleva et al., 2023; Laurençon et al., 2023) and models (Scao et al., 2022; Shliazhko et al., 2022) in the last few years, the available resources still largely lag behind English ones, hindering industrial adoption in non-English settings.

**Challenging to use at scale.** Although benefits of scaling models to enormous sizes have been amply demonstrated in terms of performance (Hoffmann et al., 2022; Wei et al., 2022b; Chowdhery et al., 2022), scale comes at a large cost in terms of hardware requirements and inference speed. Download statistics on HuggingFace show the smallest Llama model (Touvron et al., 2023a) to be the most adopted by the community, demonstrating the interest in small but capable models. LLM scaling laws demonstrate the diminishing returns of training a model of a given size past a certain amount of tokens. By continuing pre-training way past the compute-optimal threshold, performance has been shown not to saturate, enabling the training of "inference-optimal" models of high interest to the industrial and research communities (Sardana & Frankle, 2023). It is still not fully understood how model performance continues to improve at these later stages of training.

## 1.1 Contributions

In this work, we attempt to bridge the aforementioned limitations and gaps. Our main contributions can be summarized as follows.

---

[2]https://purl.stanford.edu/kh752sm9123
[3]https://www.commerce.senate.gov/2023/9/the-need-for-transparency-in-artificial-intelligence

**Contribution 1: Introduction of a highly-curated, diverse corpus in French.** We collect and openly release a 303B token corpus spanning internet data, but also literary work, speech transcripts, legal and administrative documents, scientific articles, business-related documents, etc. The corpus is distributed under permissive licenses, allowing commercial use with no restriction, and is heavily filtered, curated, and deduplicated. To our knowledge, it is the largest multi-source French language corpus released to date of sufficient quality for language modeling purposes[4].

**Contribution 2: Training CroissantLLM, a truly bilingual language model.** Nowadays, most models display some multilingual capabilities. For example, Bloom has been trained to be massively multilingual (Scao et al., 2022), Llama contains a minor proportion of non-English data in its training set (Touvron et al., 2023a) and Qwen included a significative portion of Chinese data (Bai et al., 2023a).[5] However, to our knowledge, outside of Chinese with a different alphabet (Zeng et al., 2022), no work has studied or attempted to train a multilingual model of significant scale in which English is not the dominant training language.

In this work, we train a model on a 1:1 ratio of English to French with a tokenizer optimized for bilingualism. Our end goal is to have a model less skewed towards English performance or cultural biases. We motivate this ratio by conducting careful experimentation on smaller model scales to uncover the trade-offs behind language distribution ratios. We opt for a strategy enabling a "best of both worlds" performance in these two languages, as empirically validated by scaling law experiments. Experimentally, we show the importance of integrating data from another cultural source in the acquisition of cultural knowledge, underlining the importance of the effort.

**Contribution 3: FrenchBench: A novel LLM benchmark for the French Language.** To evaluate models in French, we introduce a benchmark encompassing various tasks to assess factual knowledge, generative capabilities, language understanding, etc. This benchmark is constructed both from openly available datasets, as well as newly released manually annotated data. We evaluate and report results for our models as well as other models with French-speaking capabilities.[6][7]

**Contribution 4: Releasing high-performing, inference-optimal models for the industrial community, together with a wide range of resources for the research community.** The models we train are all released under open licenses. Our largest model is trained on a 2300:1 token to parameter ratio (115 times longer than a Chinchilla Optimal 1.3B model) leading to very strong performance in its size category. We show that model performance on downstream tasks continues to dramatically improve with lengthened pre-training runs, although model perplexity does not significantly improve. We release all checkpoints for all model sizes, as well as the exact training data seen at every step for research purposes.[8] These models are extremely efficient to run, leading to low-latency, energy-efficient on-edge inference, even on low-resource devices such as phones or personal computers. These releases[9] are motivated by a commitment to transparency to allow research and reassuring users for industrial deployment: our models comply with 81 % of criteria listed by the Foundation Model Transparency Index (Bommasani et al., 2023) (see Section 6).

The CroissantLLM initiative aims to adress the aforementionned limitations of current models through the release of an inference-optimized, small but capable model that performs well outside of English settings, and that is designed to be as open and transparent as possible. Beyond facilitating industrial LLM adoption and unlocking new generative model use cases, this project is also a proven platform for researching LLMs and kickstarting future pretraining efforts (Martins et al., 2024; Meeus et al., 2024a; Shilov et al., 2024; Meeus et al., 2024b).

---

[4]By this, we imply of sufficient quality to train a language model (little OCR errors, high-quality text) and not exclusively composed of Internet data.

[5]Qwen-VL (Bai et al., 2023b) reports 77.3 % & 22.7 % Chinese, but no information is given for the Qwen base model.

[6]Code for evaluation is available at `https://github.com/EleutherAI/lm-evaluation-harness`

[7]Another complementary initiative has been led for French model evaluation and released concurrently in Bawden et al. (2024)

[8]Training data indexing order will be released in a second stage.

[9]Code for dataset collection and filtering is available at `https://github.com/ManuelFay/llm-data-hub`. Code for model training is hosted at `https://github.com/CoderPat/croissant-llm-training`. Datasets and model checkpoints are available at `https://huggingface.co/CroissantLLM`.

|  | Size (GB) | Docs. (M) | Tokens (B) | Token/Doc | Sampling Ratio | # tokens (B) |
|---|---|---|---|---|---|---|
| French | 1258.70 | 376.27 | 303.51 | 806.63 | 4.09 | 1240.08 |
| English | 2351.13 | 591.23 | 655.64 | 1108.94 | 1.89 | 1240.09 |
| Code | 366.87 | 81.90 | 141.43 | 1726.76 | 2.04 | 288.92 |
| Parallel | 113.91 | 408.03 | 35.78 | 87.68 | 6.13 | 219.26 |
| **Total** | 4090.61 | 1457.43 | 1136.35 | 779.70 | 14.15 | 2988.35 |

Table 1: Final Data mix for CroissantLLM training

## 2 Data

The dominance of the English language in the training data of most current models is undeniable. While multilingual models like Llama leverage some non-English data (Touvron et al., 2023a), it corresponds to only a minor part of the corpus, leading to a significant drop in performance across non-English data, with noticeable "American" bias (Santurkar et al., 2023; Navigli et al., 2023). This work aims to offset this trend by using a more balanced bilingual corpus comprising English and French, as well as additional code data. Although both languages belong to the Indo-European language family, they exhibit different morpho-syntactic structures[10] and French has a richer morphology.[11] We study whether this corpus helps in reducing biases, enabling more varied knowledge sets, and unlocking non-English performance.

A variety of sources are integrated into our corpus, including carefully filtered internet data and high-quality data from a range of sources, all devoid of restrictive licenses ensuring complete openness of the data and the trained model. Data statistics are available in Table 1.[12]

The scrapping and processing code are available in our code base.[13] The license information of all datasets used is given, all allowing for permissive commercial use.

### 2.1 French Data

Table 9 lists the source and some information regarding the French corpus. Details about the data sources are expanded further below.

**Web Data.** We collect internet data from various web scraps (Oscar (Abadji et al., 2022), mC4 (Xue et al., 2021)), leveraging the CulturaX corpus (Nguyen et al., 2023) for heuristic and perplexity filtering, as well as exact and fuzzy deduplication. In total, this represents over 363 million webpages and more than 292 billion tokens, that we split using our custom tokenizer fitted on equal amounts of French and English data.[14]

We ensure data is of good quality and correctly tagged in French through sampled manual inspection and confirm French-speaking countries are well represented within the dataset. Notably, we include several news sources scrapped from Belgium, Switzerland, Canada, and Lebanon, as well as multiple African countries (Senegal, Morocco, Algeria, Cameroon, etc.) [15]

---

[10]For example, pronominal objects are placed before (resp. after) the verb in French (resp. English), both languages have different noun phrase constructions ("la commission européenne" vs. "the European commission"), etc.

[11]For example, English has 5 verb forms whereas French has 48, French has explicit inflections for grammatical genders, etc. However, note that only English adjectives have morphological constructions for expressing comparison (*e.g.* easy, easier, easiest). We refer to WALS for more details, *e.g.* `https://wals.info/feature/21B#2/26.7/152.4`

[12]As further detailed in 3.4, our data corpus contains different amounts of unique English, French, and Code tokens. We obtain our balanced training corpus by upsampling French, Code, and English data with different sampling ratios, such that no performance loss is to be expected (Muennighoff et al., 2023).

[13]`https://github.com/ManuelFay/llm-data-hub`

[14]The mC4 corpus `https://huggingface.co/datasets/allenai/c4` is released under the ODC-BY licence `https://opendatacommons.org/licenses/by/1-0/` whereas Oscar (Abadji et al., 2022) does not claim data ownership, provides an opt-out strategy for data inclusion, and filtering metadata is released under the Creative Commons CC0 license.`https://creativecommons.org/publicdomain/zero/1.0/`

[15]Contrary to certain languages such as Arabic or Portuguese in which the language differs greatly depending on the country, the written French language in these countries are largely similar with only a minor share of regionally introduced syntax and expressions

**Legal and Administrative Data.** We introduce 5.3B tokens of data from the French government's open data initiative, ranging from legal tasks to parliamentary discussions and financial transcripts (*e.g.* legal and administrative jargon). These texts originate from 13 different datasets (the OpenData corpus) and were collected from the French government's open data platform.[16] To ensure other French-speaking countries are represented, we add 68M tokens of data from Swiss legislation retrieved from government sources. We perform steps to process, filter, and run exact deduplication on these documents.

**Cultural Data.** We introduce cultural data from various sources. Notably, we retrieve all Project Gutenberg (Hart, 1971) books in the French language as of October 2023, corresponding to books released in the public domain (302 million tokens). We also download and aggressively filter manuscripts and documents from the French National Library (Bibliothèque Nationale de France), and filter for documents that belong to the public domain, have undergone an OCR process, and are of high quality.[17] To filter out low-quality OCR documents, we implement custom heuristics which we release within our code base. We run all documents through perplexity filters using KenLM 5-grams[18] fitted on the French Wikipedia split, and discard documents with perplexity values that are too high (noisy) or too low (repetitive patterns). Thresholds are set through a manual verification process. We deliberately choose to be aggressive in our filtering to ensure only high-quality data is kept and discard the largest portion of the original corpus, keeping about 27M tokens. We choose not to keep any data from the newspaper archives, as the OCR transcription is often too noisy. Additionally, we introduce famous public domain French poems custom scrapped from a French poetry website,[19] and run a set of podcasts through a high-quality speech-to-text model to obtain a textual transcription. This process is hard to scale and data splits from these sources are limited in quantity. Data from the OpenSubtitles[20] initiative is integrated, corresponding to 41.8 million tokens originating from movie subtitles.[21] Finally, we add the French data from Wikisource collected as part of the BigScience initiative (Scao et al., 2022) and obtain 2.7 billion tokens from the process.[22]

**Encyclopedia Data.** To introduce high-quality factual data to the training corpus, we integrate the French Wikipedia split from November 2023.[23] This corresponds to the latest knowledge cutoff in the training data. In total, more than 2.5 million articles are used, spanning more than 2 billion tokens.

**Industrial Data.** We scrap high-quality and publicly available data from industrial PDFs via a manually crafted list of websites, from large French and French Canadian (Quebec) companies to government agencies. This business-focused data boosts performance on a series of downstream applications related to industrial NLP. We collect over 36000 PDF multi-page documents and filter them through carefully crafted heuristics, followed by aggressive perplexity filtering.[24] In total, we obtain over 290000 documents and 191 million tokens.

## 2.2 English Data

Our English data is primarily drawn from the SlimPajama corpus (Soboleva et al., 2023), excluding copyrighted documents. Splits per data source are detailed in Table 10.

**Internet Data.** Similarly to the French dataset, we rely on carefully filtered content from an assortment of internet sources, including miscellaneous web pages and blogs. The filtering process includes heuristics and perplexity filtering, as well as large-scale deduplication (Soboleva et al., 2023). The SlimPajama corpus

---

[16]Data is released at https://echanges.dila.gouv.fr/OPENDATA/ with the ETALAB open license https://www.etalab.gouv.fr/wp-content/uploads/2017/04/ETALAB-Licence-Ouverte-v2.0.pdf

[17]Metadata is licensed under Open Etalab license https://gallica.bnf.fr/edit/und/conditions-dutilisation-des-contenus-de-gallica

[18]https://github.com/kpu/kenlm

[19]https://www.poesie-francaise.fr/

[20]https://opus.nlpl.eu/OpenSubtitles2016.php

[21]https://www.opensubtitles.org

[22]Licensed under CC BY-SA 4.0, https://en.wikisource.org/wiki/Wikisource:Copyright_policy

[23]https://huggingface.co/datasets/wikimedia/wikipedia with a CC-By-SA 3.0 license

[24]Data is public internet data that we release under MIT license with an opt-out scheme https://huggingface.co/datasets/manu/illuin_layout_dataset_text_only

includes internet data from the CommonCrawl[25] and C4[26] web scraps, as well as data sourced from Github textual content[27] and the StackExchange forums.[28]

**Miscellaneous.** Other non-internet-based data sources are included in the SlimPajama dataset, such as scientific articles from Arxiv[29] and English documents from Wikipedia.[30] The SlimPajama dataset is also comprised of the "Books" subcorpus, obtained by downloading all book documents from Bibliotik.[31] Some of the documents within this last corpora have been flagged by their owner as proprietary data. We filter out all documents from this subcorpus, and replace them with data from the open-source Project Gutenberg (Hart, 1971) English books under public domains.[32]

**Gutenberg Canaries.** To assess model memorization to inform about the risks of including private or sensitive data within the training set, we stress test the model by including "canaries" (Carlini et al., 2019). These correspond to samples that have been intentionally modified and/or repeated and included within the model training set, and that will enable a posteriori evaluation of the model capacity to memorize data in a "worse than worst-case" situation.[33] In total the canaries represent 555 million tokens, representing less than 0.04 % of the total tokens seen during training.

## 2.3 Code Data

In line with most recent models (Chowdhery et al., 2022; Scao et al., 2022; Touvron et al., 2023a), we integrate code data into our training corpus. Notably, previous work shows that code data benefits natural language tasks and can be particularly useful in data-constrained settings (Muennighoff et al., 2023). Therefore, we include 140B tokens of code data in several common programming languages. Splits and number of tokens are detailed in Table 11.

**The Stack & StarCoder.** We rely on the efforts of the StarCoder project (Li et al., 2023a), and use their high-quality filtered code data from The Stack corpus (Kocetkov et al., 2022).[34] We keep only high-resource programming languages (Java, Javascript, Python, C, C++, SQL) and Jupyter[35] notebooks, as well as a few samples of formatted data (JSON, Tex) and scripting languages (shell scripts, Dockerfiles).[36]

**Extra Python code.** We extend the corpus with several other sources of Python code due to the popularity of the language in the community. Firstly, we add Pypi packages from recent code dumps,[37] that are filtered to keep only Python and Jupyter files.[38] Secondly, in order to promote high-quality problem-solving-centered code, we integrate 1.2B tokens of Python3 data from competitive coding contests (Li et al., 2022).[39] Lastly, following the success of learning from textbooks (Li et al., 2023b), we add commented Python code constructed by combining code and text cells from Jupyter Notebooks through the CodeParrot initiative.[40]

## 2.4 Parallel Data

Following previous work (Anil et al., 2023), we incorporate vast quantities of parallel data, in our case high-quality English-French translation pairs, in order to improve the multilingual capabilities of the model (Briakou et al., 2023).

---

[25]Common Crawl license `https://commoncrawl.org/terms-of-use`

[26]ODC-BY license

[27]Only content under MIT, BSD, and Apache licenses are kept

[28]CC-By-SA 4.0 license `https://archive.org/details/stackexchange`

[29]`https://arxiv.org/`, with author opt-out options

[30]CC-By-SA 3.0 license

[31]`https://huggingface.co/datasets/the_pile_books3`

[32]From `https://huggingface.co/datasets/pg19` with an Apache 2.0 license

[33]This work is led in parallel to the CroissantLLM project and findings will be independently published.

[34]Data are drawn from various sources and falls under 193 different permissive licenses. We use the version 1.2 of the corpus, which has been filtered with respect to data owners opt-out option information.

[35]`https://jupyter.org/`

[36]`https://docs.docker.com/engine/reference/builder/`

[37]`https://py-code.org/datasets` from permissive licensed code

[38]`https://huggingface.co/datasets/vikp/pypi_clean`

[39]Under CC-By-4.0 license

[40]`https://huggingface.co/datasets/codeparrot/github-jupyter-code-to-text` Apache 2.0 license

| Model | Params (M) | Layers | Hidden size | Inter. size | KV heads |
|-------|-----------|--------|-------------|-------------|----------|
| XXS | 100.7 | 6 | 1024 | 4096 | 8 |
| XS | 202.5 | 12 | 1024 | 4128 | 8 |
| S | 341.5 | 12 | 1536 | 4128 | 12 |
| Base | 1214.3 | 24 | 2048 | 5504 | 16 |

Table 2: Model information for scaling laws. Parameter count excludes embedding and output parameters.

**Opus.** We extract subsets of sentence pairs spanning multiple domains from the OPUS corpus (Tiedemann, 2012).[41] Statistics are described in Table 10. In total, we include 400 million parallel sentences and about 36 billion tokens. The data is filtered through a rigorous cleaning pipeline: (1) BiFixer (Ramírez-Sánchez et al., 2020)[42] is first used to remove duplicate data through fuzzy matching techniques; (2) BiCleaner[43] is then used to filter data using heuristic and perplexity filters; (3) finally, the state-of-the-art NMT quality estimator CometKiwi (Rei et al., 2022b) is used to keep only top quality translation pairs.

**Theses.** To incorporate versatile academic and scientific language, we augment our dataset with French theses abstracts along with their author-generated English translations. This corresponds to 95000 documents and more than 80 million high-quality tokens.[44]

**Song Lyrics.** Our dataset integrates song lyrics in both French and English, scrapped from a specialized community-driven lyrics translation website.[45] As such, our model is trained with radically different linguistic styles (*e.g.* colloquialism), and the wide range of topics can help the model to capture cultural nuances. Lyrics have been translated by hand by the website community. With a total of 70k songs, we have built up a corpus of 53M tokens covering different periods (80s, 90s, 20s, etc.) and musical genres (rap, rock, jazz, etc.) To preserve colloquial expressions and cultural subtleties, we have not filtered song lyrics for explicit content. We validate the original language metadata of the songs through Google's language-detection algorithm.

## 3 Training

Our main goal was to train a high-performing, yet resource-friendly bilingual model while optimizing performances across both languages. To focus on the specific challenges of the bilingual paradigm, we rely on previous work to motivate many of our design and hyperparameter choices (Touvron et al., 2023b).

### 3.1 Model Architecture

We use the Llama architecture (Touvron et al., 2023a), a decoder-based transformer, trained with rotary position encodings (Su et al., 2023) and a context length of 2048. We construct 4 different model sizes by jointly scaling the number of attention heads, hidden size, and hidden layers. Table 2 summarizes the sizes of each model in the family.

### 3.2 Tokenizer

Most LLM tokenizers are fitted on English-centric corpora with an information-theoretic optimization objective, for example, Byte-Pair encoding (Sennrich et al., 2016) or Unigram (Kudo, 2018), leading to good fertility[46] values (low token per word ratio) on English text, but high fertility in other languages. These

---

[41]Free license `https://opus.nlpl.eu/`

[42]`https://github.com/bitextor/bifixer`

[43]`https://github.com/bitextor/bicleaner`

[44]Data licensed under Etalab open license `https://www.etalab.gouv.fr/wp-content/uploads/2017/04/ETALAB-Licence-Ouverte-v2.0.pdf`

[45]`https://www.lacoccinelle.net`, with opt-out options for data owners and community-contributed translations under free-use license.

[46]See Appendix E for a definition.

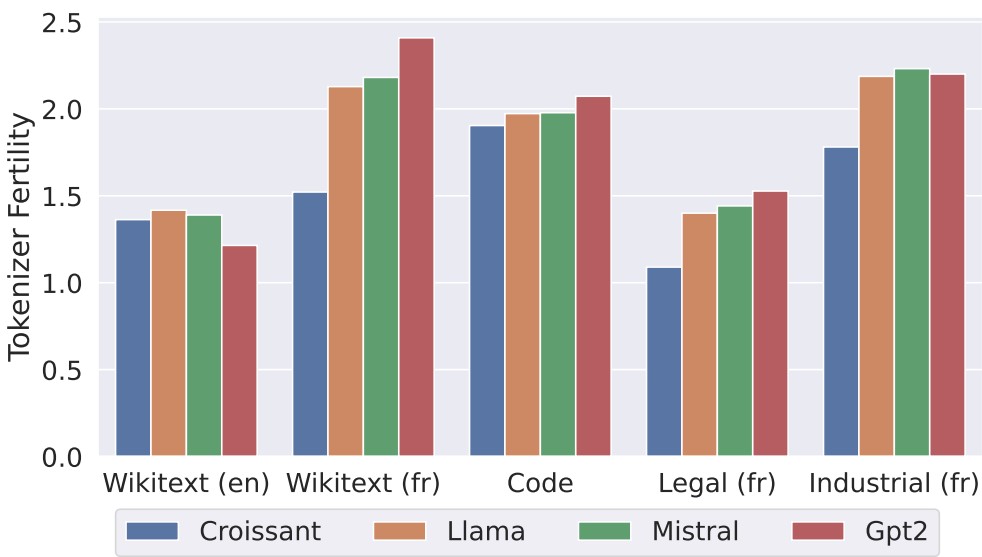

Figure 2: Fertility on unseen test sets using various tokenizers. Lower is better.

phenomena make processing in other languages slower and more costly (Rust et al., 2021). Furthermore, subword splits in non-English languages mechanically carry less semantical meaning, potentially being a factor in the degraded performance of models on non-English languages (Rust et al., 2021).

**Tokenizer training.** We fit our CroissantLLM tokenizer on a corpus of 100B high-quality tokens, with splits of English, French, and code data. We use SentencePiece[47] to train a Byte-Pair Encoding tokenizer with a vocabulary size of 32000 tokens, 100 special placeholder tokens, whitespace separation, and byte fallback, inspired by Touvron et al. (2023a); Jiang et al. (2023). The data corpus used to fit the tokenizer is made available,[48] and notably contains large amounts of French data to skew the vocabulary construction process towards optimizing for French as well.

**Improved fertility rates.** The focus on English, French, and Code enables the CroissantLLM tokenizer to display smaller fertility rates on French texts than the Mistral and Llama models with similar vocabulary sizes, all the while also displaying slightly smaller rates than both in English and Code (Figure 2). This is due to the multilingual support of both Llama and Mistral tokenizers which need to allocate some vocabulary tokens to frequent character patterns from other languages. Roughly, the Llama tokenizer is 17 % less token efficient at encoding French internet data, and up to 40 % less efficient on clean encyclopedia French texts, implying that the 303B unique French tokens in our data training set correspond to more than 360B tokens with the Llama tokenizer. This enables us to pack more data in fewer tokens, leading to improvements in training and inference efficiency.

### 3.3 Selecting an optimal language ratio

A crucial question when training a bilingual model is how to effectively *weight* data from the two languages to achieve a good trade-off between performance in both. While, intuitively, training on an equal mixture of English and French data may seem to be the obvious solution, differences in data quality available for each language coupled with transfer learning dynamics between both languages could imply that a balanced mix might be sub-optimal. However, training multiple models with different data mixes for comparison is prohibitively expensive.

To offset this computational cost, we leverage recent findings on scaling laws (Kaplan et al., 2020b) that show that we can predict the performance of our model by training smaller models on the same dataset.

---

[47]https://github.com/google/sentencepiece
[48]https://huggingface.co/datasets/manu/tok-corpus-shuffled

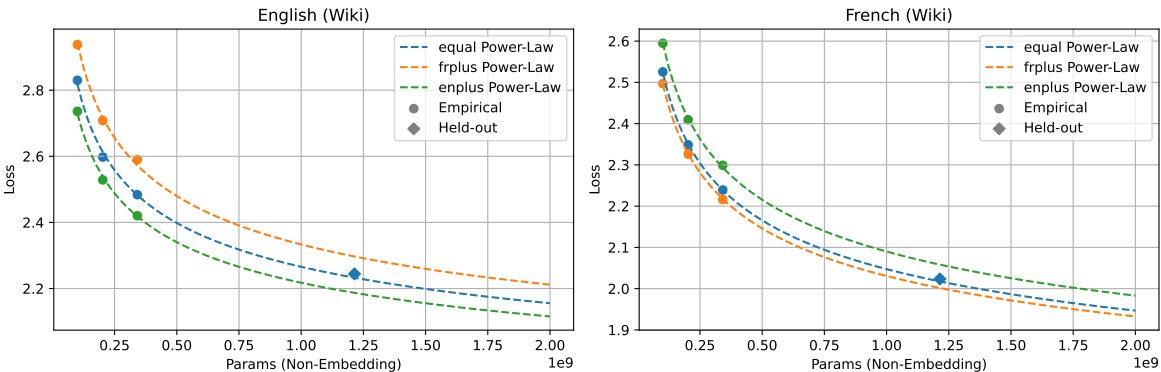

Figure 3: Evolution of test cross-entropy loss with model size in English (left) and French (right), for the *wiki* domain, as well as the fitted joint scaling laws,

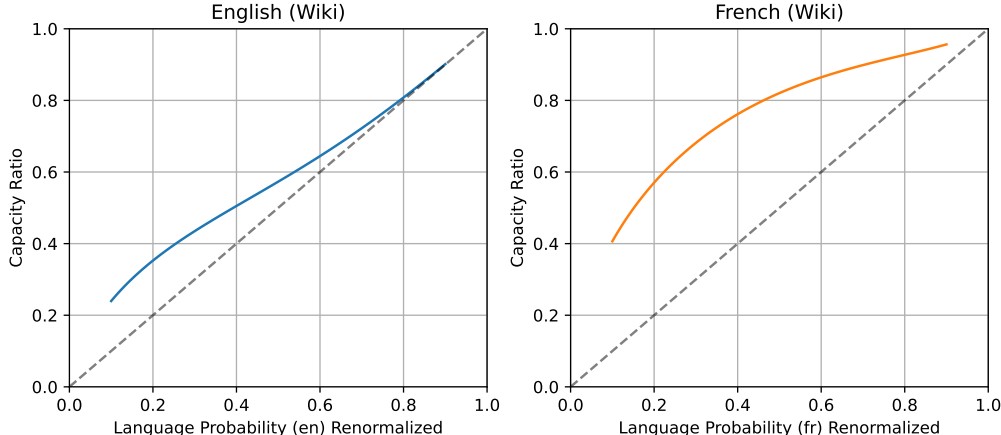

Figure 4: *Effective capacity ratio* (as predicted by our fitted joint scaling law) for English and French as we change the weight of each language.

In particular, Fernandes et al. (2023) found that, for *multilingual* models, by training smaller models with varying weights for each language in the data mix, one can fit a *multilingual, joint* scaling law that can predict the language performance trade-off of larger models, even for novel language weightings not encountered during the fitting of the scaling law.

As such, we fit a joint scaling law as described by Fernandes et al. (2023) for each language, by training 3 smaller model sizes on 3 different data mixes with varied ratios of English and French data (keeping the amount of Code data fixed). The corpus for these scaling law experiments is a subsampled variant of the larger corpus and is detailed in Appendix B.5. We define 3 data mixes by varying the language sampling ratio: (1) **equal** containing 40 % English data, 40 % French data and 20 % Code data; (2) **frplus** containing 20 % English data, 60 % French data and 20% Code data; and (3) **enplus** containing 60 % English data, 20 % French data and 20 % Code data. We then trained a 1.3B model on these subsets of the data for one of the data mixes to validate their predictive power.

Figure 3 shows the performance predicted by jointly-fitted scaling laws as we scale the model and vary the language weightings on the Wiki data validation split. First, we see that the fitted scaling law is able to predict the performance of the larger model almost perfectly. Secondly, changing the weight of each language in training has a non-symmetrical impact on language performance: by increasing the (relative) weight of French from 50 % to 75 %, we get a marginal performance increase in French, while performance in English

drops significantly. This fact is made clear by plotting the *effective capacity ratio*[49] of each language as we change the language weight (Figure 4): the "gains" in parameters from increasing weight of French data are minimal past the 50 % mark.

These findings showcase that **multilinguality comes at a price**, and training a bilingual model implies accepting a performance loss on a target language compared to an equivalent model trained on a monolingual corpus.

We find equal ratios of English and French data lead to minimized performance hits across both languages (Figure 3) and opt to train our base model in this data configuration.

### 3.4   Final data distribution

Our final dataset is composed of 1.1T unique tokens that originate from sources of various languages, qualities, and quantities. To craft a training set with a language and data distribution that suits our objectives, we upsample some of the sources, notably to balance out French and English data and increase the share of parallel data in our training run. Following work by Muennighoff et al. (2023) and Luukkonen et al. (2023) on data-constrained language modeling scaling laws, we upsample French text by a factor of two, and parallel data by a factor of 3. For a 3T token training run, this enables the model to see French data at most 4 times, and English and code data twice, which should have negligible impacts on the performance (Muennighoff et al., 2022). The final data distribution is shown in Table 1.

All data is provided from the above-listed sources and no synthetic or augmented data is used. Data licenses and copyright information are given for every split to the best of our ability. The data collection and filtering process to construct our final mix from the above-listed sources is entirely done by the authors of this paper, who are employed by the universities or private companies described through their affiliations, under their countries' data protection laws, and compensated at market rates or following the academic salary grid of their institution.

### 3.5   Training framework

We train our models on a modified version of Megatron-Deepspeed,[50] a training framework built on top of PyTorch. Training is done on a dedicated Nvidia A100 SXM4 80 Gb supercomputer partition with 30 octo-GPU nodes. We rely on the HuggingFace Transformers and Datasets library for model and data manipulation.

To maximize efficiency, we set the micro-batch size per device to 8 sequences of length 2048, and use 4 gradient accumulation steps, resulting in a total batch size of $8 \times 4 \times 30 \times 8 = 7680$ samples, or $7680 * 2048 = 15,728,640$ tokens. We achieve a mean efficiency of around 120 TFLOP[51] per second with activation checkpointing, leading to a total compute estimate of $4.30e22$ FLOPS. Standard Cross-Entropy losses are used on a Causal Language Modeling objective.

### 3.6   Training losses

Training lasts 17 days for a total of 99648 GPU hours, and we chose not to manually intervene, letting the model recover on its own after occasional loss spikes. [52] We train with a max learning rate of $3e-4$, 1000 warmup steps, and a cosine learning rate with a minimum value of $1e-5$. Curves suggest the model still

---

[49]The relative monolingual model size required to match the performance of the multilingual model on a language. See E for more details

[50]https://github.com/deep-spin/Megatron-DeepSpeed

[51]160 TFLOP per second for our scaling law experiments with one GPU node only

[52]Previous literature rarely mentions handling of such issues. Some models have reloaded training by skipping batches (Zhang et al., 2022; Anil et al., 2023), others have seemed to ignore it if loss recovered (Touvron et al., 2023a). In our case, we used gradient clipping, weight decay and tuned optimizer values (Zhang et al., 2022; Touvron et al., 2023b), to minimize the occurrences of such loss spikes. . . During training, as spikes were few and far between, and loss seemed to recover quite rapidly every time and without visible consequences, we opted to let the model train with no intervention. We highlight that since our checkpoints and dataloaders are openly available, this enables future experimentations around these loss spikes.

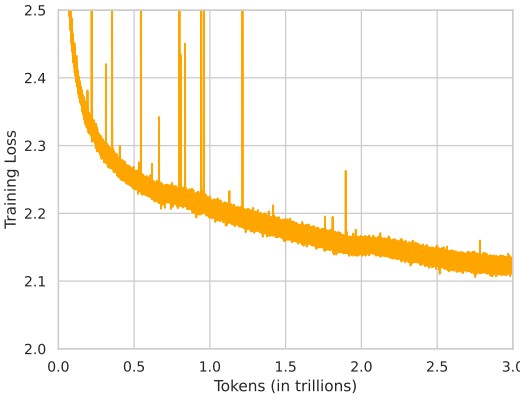 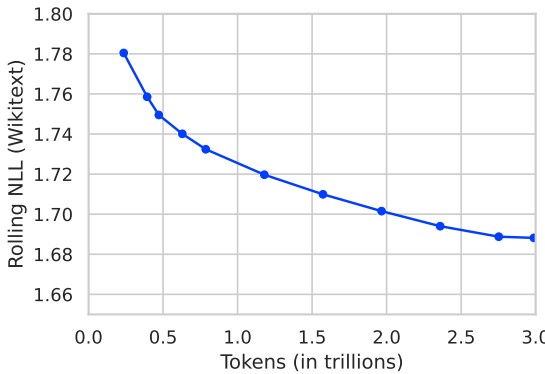

Figure 5: (Left) Training loss with respect to the number of seen tokens. (Right) Validation perplexity (Averaged Log Likelihood) on Wikitext (English), computed with a rolling stride

has not reached a performance plateau after 3T tokens (Figure 5). Checkpoints are stored every 5k steps and released with the rest of the project artifacts.

### 3.7 Environmental impact

The model was exclusively trained on a supercomputer operating on low-carbon nuclear electricity. Between experimental runs, scaling laws, and the final training, 123k A100 hours were used. The Thermal Design Power of the NVIDIA A100 SXM4 80Gb used is 400W corresponding to a total power consumption of 49.2 MWH and considering a grid carbon intensity of 57 gCO2eq/kWh, we estimate a carbon footprint of 2.80 tons of CO2 (Luccioni et al., 2022) emitted during training.[53]

Interestingly, the model we trained is not "compute-optimal" according to Chinchilla laws (Hoffmann et al., 2022), meaning that less computing could have been used to train a larger model with the same performance. However, our model aims to be used for inference purposes at industrial scales. Our training paradigm is thus to absorb the downstream inference costs, by training a smaller model on a lot more tokens to obtain an inference-optimized model equivalent in performance to a bigger compute-optimal model (Sardana & Frankle, 2023). Each inference of the final model is thus vastly more energy-efficient than a Chinchilla optimal model of equivalent performance ($> 3B$ parameters), and can even run on CPU or mobile devices. Relying on estimates of Kaplan et al. (2020a), at inference, CroissantLLM represents roughly 2.6 GFLOPS per token.

## 4 Evaluation Benchmarks

We hope to extend base model evaluation past English benchmarking alone and assess model capabilities in French, aiming for broad coverage across orthogonal capabilities to observe the effect of truly bilingual pre-training. Our evaluation efforts are rooted in transparency, and all results reported in the main technical report are reproducible through code that is open-sourced and public data.[54]

### 4.1 English

In English, we evaluate on standard LLM evaluation benchmarks.

---

[53]Factoring in an intentionally large data center Power Usage Effectiveness of 1.2 (Luccioni et al., 2022), we estimate an emission of 3.36 tons of CO2.

[54]All benchmarks are made available and links will be provided in the non-anonymous version of this work.

**HellaSwag.** HellaSwag (Zellers et al., 2019) is a dataset specifically crafted to challenge common-sense reasoning abilities of models by requiring them to predict the endings of sentences in a way that relies on information not present in the preceding context. It focuses on capturing a nuanced and context-dependent understanding of language.

**PiQA.** PIQA is a dataset for common-sense reasoning and was created to investigate the physical knowledge of existing NLP models (Bisk et al., 2019).

**SciQ.** The SciQ dataset contains 13,679 crowdsourced science exam questions about Physics, Chemistry, and Biology, among others. The questions are in multiple-choice format with 4 answer options each (Welbl et al., 2017).

**Arc-C.** The AI2 reasoning challenge dataset (Clark et al., 2018) consists of 7,787 authentic grade-school level, multiple-choice science questions, designed to stimulate research in advanced question-answering. The dataset is divided into a Challenge Set and an Easy Set, with the Challenge Set comprising questions that were answered incorrectly by both a retrieval-based algorithm and a word co-occurrence algorithm. Additionally, the dataset includes a corpus of over 14 million science sentences relevant to the task and provides three neural baseline models.

**MT-Bench.** MT-Bench (Zheng et al., 2023) contains a set of prompts designed to evaluate models on their multi-turn conversation and instruction-following abilities, covering various core model abilities; writing, roleplay, extraction, reasoning, math, coding, knowledge I (STEM), and knowledge II (humanities/social science). MT-Bench performance has been shown to best correlate with human-rated appreciation of a model through the LM-Sys model arena.

## 4.2 French

We aim to evaluate models on their capabilities in French, along several axes including vocabulary, grammar, reading comprehension, factual knowledge, biases, and generative capacities, etc. To this end, we introduce *FrenchBench*, a novel LLM evaluation benchmark for the French language, testing a large array of model skills in various settings.

FrenchBench comprises several tasks, some included from previous benchmark datasets, others newly released with this work.

### 4.2.1 FrenchBench Gen

FrenchBench assesses the generative capabilities of LLMs in a few-shot setting. Tasks include title generation, summarization, question generation, and question answering. We detail the tasks and the evaluation metrics used below.

**FQuaD.** FQuaD (d'Hoffschmidt et al., 2020) is a French Question Answering dataset, containing manually annotated sets of Wikipedia passages, questions, and extractive answer spans in the Squad format. This high-quality dataset is one of the rare human-annotated French datasets and we rely on its public evaluation split for 4 of the FrenchBench tasks.

*FQuADGenQ* is a question generation task in which passages and answers are given to the model in a few-shot manner, and we compute the ROUGE1 score (Lin, 2004) with the gold questions.

*FquadGenAns* is the classic question-answering task, but models generate the answer span themselves, and the ROUGE1 score is computed with the gold extractive answer span.

*MultiFQuAD* [55] is a FQuAD variant, with a publicly released evaluation set, in which answers can consist of multiple disjoint spans. We evaluate performance on the concatenation of these gold extractive spans using the ROUGE1 score.

**French Trivia.** The French Trivia dataset is built from online trivia questions pertaining to French culture. Answers are short and meant to assess latent model knowledge and the impact of pre-training data and

---

[55]Previously unreleased dataset, evaluation set is released under CC-By-NC SA 4.0 license with this work

cultural references. Intentionally, questions are formulated in English for comparison with monolingual English models.[56]

### 4.2.2 FrenchBench Multiple Choice

We also assess reasoning, factual knowledge, linguistic capabilities, and model biases through a series of few-shot classification tasks, on which models are given multiple completions (multiple choice), and the answer with the highest likelihood is selected. We experimented with multiple question templates. In the MMLU format, the multiple potential answers are given after the question prefixed by a letter (A, B, C, D) and the model must guess the correct answer by predicting the correct answer's letter. In HellaSwag formatting, the model must complete the sentence and the model chooses the most likely continuation sequence, without prior knowledge of all other options. We find HellaSwag formatting is less abstract, and enables smaller size models to perform better.

**French Language Test.** The French Language Test is a dataset crafted to assess the grammar and vocabulary capabilities of models through language tests. It provides a structured evaluation of a model's linguistic proficiency, aiming to measure its competency in understanding and generating coherent and grammatically accurate sentences in the French language. It is composed of a *fr-grammar* and *fr-vocabulary* multiple choice test.

**French Hellaswag and Arc-C.** These datasets correspond to machine translations made by GPT3.5 of HellaSwag and Arc-C to French.[57] Manual verification of the translation quality indicates the translations to be far from perfect but sufficient for these datasets to act as a correct performance proxy.

**OrangeSum.** OrangeSum[58] (Eddine et al., 2020) is a summarization dataset constructed from online News articles. Two standard French summarization tasks span from this dataset; *OSum(T)* in which the model is tasked with generating the title from the article body, and *OSum(A)* in which the model must generate the first paragraph of the article aimed to be an abstract of the article. We select the abstract generation task, and measure performance with the ROUGE1 score.

### 4.3 Other Tasks

**MT-Bench French.** Mt-Bench French[59] is a translated and adapted version of MT-Bench in French with all questions having undergone rigorous human review and adaption to guarantee authentic wording, and coherence, and to account for cultural discrepancies.

**Translation.** Translation capabilities are evaluated through the test set of the 2014 WMT French-English and English-French tasks (Alves et al., 2023). We measure performance using BLEU score (sacreBLEU, Papineni et al., 2002; Post, 2018), and COMET (Rei et al., 2022a). We also report FLORES (Team et al., 2022) and TICO (Anastasopoulos et al., 2020) scores.

**Belebele.** Belebele is a challenging reading comprehension dataset, with multiple choices, released across 122 languages in parallel format (Bandarkar et al., 2023). We leverage the English and French splits.

## 5 Benchmark results

**Baseline models.** To evaluate CroissantLLM, we compare with an array of various models, varying in parameter size, pre-training language distribution, training corpus size, etc.

For "monolingual" English models, we evaluate Pythia-1.4B (Biderman et al., 2023) trained on 300B tokens, OPT-1.3B (Zhang et al., 2022) trained on 180B tokens, and TinyLlama(1.1B) (Zhang et al., 2024). TinyLlama is a very strong English baseline, as it holds many similarities to CroissantLLM. It is a 1.1B model trained on 3 trillion tokens with the same English corpus as the Croissant base. Although it contains some

---

[56]This is a previously unreleased dataset, released under MIT license with this work.
[57]https://github.com/laiviet/lm-evaluation-harness/tree/main/datasets
[58]https://huggingface.co/datasets/orange_sum
[59]https://huggingface.co/datasets/bofenghuang/mt-bench-french

| Task | Arc-e | Belebele (eng) | Hellaswag | PiQA | SciQ | Avg |
|---|---|---|---|---|---|---|
| GPT-fr(1B) | 0.27 | 0.28 | 0.29 | 0.54 | 0.68 | 0.41 |
| Pagnol-XL(1.5B) | 0.34 | 0.25 | 0.31 | 0.56 | 0.76 | 0.44 |
| mGPT(1.3B) | 0.48 | 0.23 | 0.35 | 0.66 | 0.62 | 0.47 |
| Bloom(1.1B) | 0.55 | 0.24 | 0.36 | 0.68 | 0.89 | 0.54 |
| OPT(1.3B) | 0.61 | 0.23 | 0.42 | 0.72 | 0.92 | 0.58 |
| Pythia(1.4b) | 0.63 | 0.25 | 0.42 | 0.71 | 0.92 | 0.59 |
| Bloom(3B) | 0.64 | 0.24 | 0.42 | 0.71 | 0.93 | 0.59 |
| **CroissantLLM**(1.3B) | 0.62 | 0.28 | 0.42 | 0.72 | 0.92 | 0.59 |
| **CroissantCool**(1.3B) | 0.62 | 0.26 | 0.43 | 0.73 | 0.92 | 0.59 |
| TinyLlama(1.1B) | 0.65 | 0.26 | 0.45 | 0.73 | 0.94 | 0.61 |
| Llama2(7B) | 0.79 | 0.46 | 0.56 | 0.79 | 0.97 | 0.72 |
| Mistral(7B) | 0.83 | 0.85 | 0.60 | 0.82 | 0.98 | 0.81 |

Table 3: English Benchmarks (5-shot results)

amount of high-quality non-English data, it is only a minor share of the training corpus, the main data sources being English and code data. As such, it trains on much more English tokens than CroissantLLM. All models are trained way past Chinchilla optimality ($\sim$26B tokens for a 1.3B model).

For monolingual French models, we use GPT-fr (Simoulin & Crabbé, 2021), a 1B model trained on 16.3B tokens, as well as the PagnolXL(1.5B) model (Launay et al., 2021), both in their author submitted HuggingFace implementations.

We also compare CroissantLLM with multilingual models, notably Llama2(7B) (Touvron et al., 2023b) trained on 2T tokens, Mistral7B (Jiang et al., 2023), and Bloom (Scao et al., 2022) models (from 1.1B to 3B), trained on 350B tokens each. We note that although the largest Bloom model is undertrained according to Chinchilla optimality (Hoffmann et al., 2022), smaller models are trained on the same number of tokens, making them largely more inference optimal and thus strong contenders. Finally, in the same size category, we evaluate mGPT (Shliazhko et al., 2022) a 1.3B model trained on 440B tokens.

Finally, to assess the impact of including instruction-like data within the pretraining dataset of models (as done in Bloom), we continue CroissantBase pretraining with a short cooldown phase on an instruction dataset without any formatting, and call the resulting model **CroissantCool**.

## 5.1 Base model

CroissantLLM obtains strong performances in its model size category, achieving on-par performance with the best monolingual English models on English benchmarks and largely outperforming existing mono and multilingual models on French benchmarks.

**English.** On English benchmarks (Table 3), CroissantLLM displays performances almost equivalent to those of TinyLlama, which has trained on much more English data. We see training on such a large quantity of English tokens enables our model to edge out similarly sized monolingual models trained on fewer tokens (OPT, Pythia), and larger multiingual models (Bloom 3B) demonstrating the interest of pursuing training past Chinchilla optimality, especially when splitting model capacity across languages.

**French.** On French classification benchmarks, CroissantLLM largely outperforms models of similar sizes trained on mostly monolingual English or French data, and multilingual models (Table 4). Performance is on par with the Bloom(3B) model, which is about 3 times as large. An interesting phenomenon can be noticed, especially on generative benchmarks assessed in few-shot settings: "base" models trained with instruction-like data perform a lot better. This is noticeable with the Bloom(3B) model which outperforms the otherwise vastly superior Llama2(7B) model on several tasks, or through the performance gains of CroissantCool with respect to CroissantBase.

| Task | Hellaswag(fr) | Arc-c(fr) | fr-vocab | fr-grammar | Belebele(fr) | Avg |
|---|---|---|---|---|---|---|
| OPT(1.3B) | 0.28 | 0.19 | 0.50 | 0.61 | 0.28 | 0.37 |
| Pythia(1.4B) | 0.30 | 0.20 | 0.61 | 0.76 | 0.23 | 0.42 |
| TinyLlama(1.1B) | 0.33 | 0.23 | 0.64 | 0.67 | 0.25 | 0.42 |
| mGPT(1.3B) | 0.27 | 0.20 | 0.71 | 0.73 | 0.23 | 0.43 |
| GPT-fr(1B) | 0.30 | 0.19 | 0.70 | 0.79 | 0.24 | 0.44 |
| Bloom(1.1B) | 0.34 | 0.22 | 0.76 | 0.79 | 0.24 | 0.47 |
| Pagnol-XL(1.5B) | 0.33 | 0.21 | 0.77 | 0.82 | 0.27 | 0.48 |
| **CroissantCool**(1.3B) | 0.40 | 0.26 | 0.77 | 0.78 | 0.23 | 0.49 |
| **CroissantLLM**(1.3B) | 0.40 | 0.26 | 0.75 | 0.80 | 0.27 | 0.50 |
| Bloom(3B) | 0.40 | 0.27 | 0.78 | 0.81 | 0.23 | 0.50 |
| Llama2(7B) | 0.44 | 0.38 | 0.76 | 0.77 | 0.43 | 0.56 |
| Mistral(7B) | 0.49 | 0.47 | 0.78 | 0.78 | 0.78 | 0.66 |

Table 4: FrenchBench MC (5-shot results)

| Task | FGenQ | FGenAns | MultiFQuAD | OSum(A) | FTrivia | Avg |
|---|---|---|---|---|---|---|
| Pagnol-XL(1.5B) | 0.06 | 0.04 | 0.03 | 0.03 | - | *0.04 |
| GPT-fr(1B) | 0.04 | 0.02 | 0.05 | 0.11 | - | *0.06 |
| mGPT(1.3B) | 0.01 | 0.00 | 0.02 | 0.03 | 0.33 | 0.08 |
| OPT(1.3B) | 0.09 | 0.18 | 0.21 | 0.17 | 0.39 | 0.21 |
| Bloom(1.1B) | 0.17 | 0.28 | 0.26 | 0.10 | 0.31 | 0.23 |
| Pythia(1.4B) | 0.15 | 0.34 | 0.27 | 0.21 | 0.44 | 0.28 |
| **CroissantLLM**(1.3B) | 0.19 | 0.40 | 0.33 | 0.10 | 0.52 | 0.31 |
| Bloom(3B) | 0.21 | 0.47 | 0.37 | 0.18 | 0.47 | 0.34 |
| TinyLlama(1.1B) | 0.18 | 0.46 | 0.41 | 0.23 | 0.45 | 0.35 |
| **CroissantCool**(1.3B) | 0.20 | 0.45 | 0.36 | 0.27 | 0.53 | 0.36 |
| Llama2(7B) | 0.25 | 0.68 | 0.60 | 0.30 | 0.70 | 0.50 |
| Mistral(7B) | 0.33 | 0.78 | 0.64 | 0.31 | 0.74 | 0.56 |

Table 5: FrenchBench Gen (5-shot ROUGE1 results). Bloom models seem to have strong performance on QA tasks (Fquad), likely due to the inclusion of Question Answering datasets in its pretraining corpus (Laurençon et al., 2023). Pagnol-XL and GPT-fr are trained exclusively on French text and as such cannot be fairly evaluated on the French Trivia test.

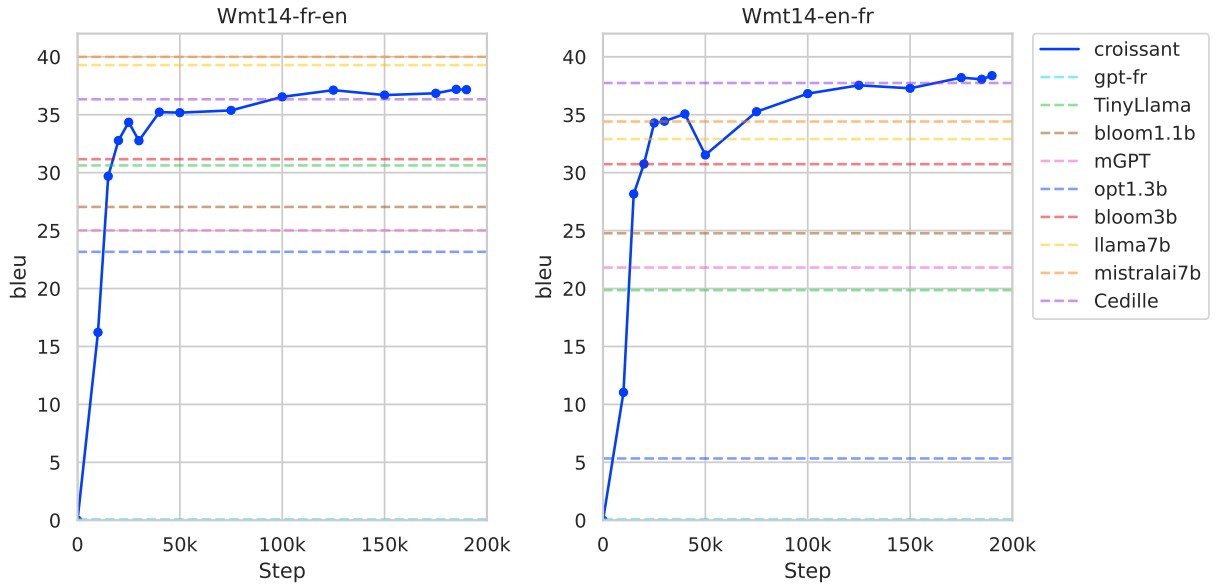

Figure 6: Performance evolution on the WMT Translation task (5-shot)

**Improvements throughout training.** The model performance continues to improve on downstream tasks during the entirety of training. We report WMT14 translation performance in Figure 6, and observe similar trends across all tasks. The benefits of training past Chinchilla optimality are clear, and although there are diminishing returns past a certain number of steps, training does not seem to saturate. In low training step settings, performance appears to emerge suddenly, reflecting emergent performance experiments in the literature most often obtained through model scaling (Wei et al., 2022b).

**French Trivia.** One main question this work attempts to tackle is whether training on bilingual data goes beyond augmenting the language understanding and writing capabilities of a model in another language, but also equips the models with novel knowledge and different cultural biases. We evaluate French cultural knowledge on a Trivia task, consisting of questions about France-related topics, asked in English (Table 5), and score results obtained in 5-shot settings with ROUGE-1.[60] CroissantLLM is the best performing model evaluated under the 7B size, outperforming English-centric models by significant margins. This knowledge gap showcases the effect of the pretraining data mix in specific knowledge acquisition, underlining the interest of integrating vast amounts of varied and multilingual data when training models aiming for broad knowledge coverage.

**Overall.** The 1.3B CroissantLLM displays top-of-its-class performance across both languages and all benchmarks, even edging out larger models such as Bloom(3B) on most tasks. All models remain far off from the performance of the strong 7B Llama and Mistral models.

## 5.2 Finetuning

Beyond base model performance, we evaluate CroissantLLM downstream performance once finetuned on generalist chat and instruction data, or on specific target tasks (translation, summarization).

---

[60]As heuristic-based metrics are often insufficient to capture the diversity of possible answers (Faysse et al., 2023), we also score predictions using GPT4 as a judge, and confirm the ROUGE1 metric is well suited for this task given the closed and short nature of the answers.

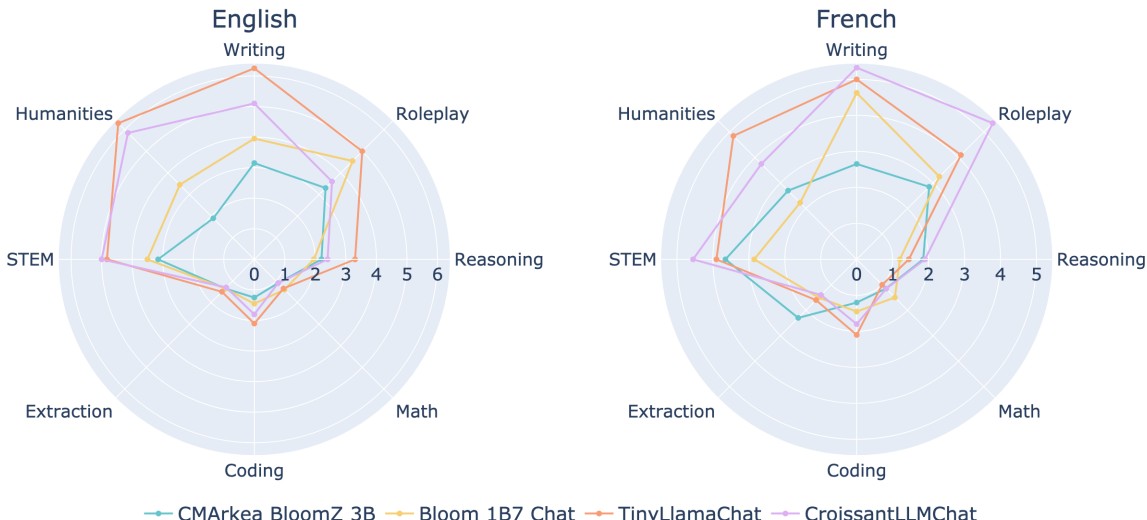

Figure 7: MT Bench Results (Both Turns)

### 5.2.1 Chat Model

It has been shown that supervised fine-tuning on instruction or chat datasets enables leveraging model capabilities to their fullest (Wei et al., 2022a).

**Training.** We finetune the base model on public Chat datasets Ultrachat (Ding et al., 2023) and Wildchat (Zhao et al., 2024) containing ChatGPT interactions in English and French. We also incorporate 12k samples of translation data (4 % of the SFT dataset). We run finetuning on CroissantLLM, as well as the Bloom-1b7 and TinyLlama models for comparison. The obtained models are further suffixed with "Chat".

**MT-Bench.** We evaluate models on the MT-Bench benchmarks, both in English and French. Although a large difference in performance can be noted between the Bloom model and Croissant in favor of the latter, performance differences with TinyLlama are not as significant, neither in English nor in French. CroissantLLMChat performs strongly in open-ended writing categories (writing, roleplay, humanities) but struggles with reasoning and extractive tasks. Turn 2 performance (reformulation under constraints) is largely lower than Turn 1 performance as can be seen in Figures 11 and 12. Our CroissantLLMChat model also vastly outperforms the BloomZ 3B model trained by CMArkea on a large chat finetuning corpus (Delestre, 2023).

This hints at the fact that quasi-monolingual models with only a minor share of another language in their pretraining corpus can be adapted to a reasonable extent, through subsequent finetuning or continued pre-training, although large pre-training corpora are necessary to incorporate sufficient knowledge and reasoning abilities within the base models. We notice large correlations between generation temperature and performance and find CroissantLLMChat works a lot better with higher temperatures ($\geq 0.4$). For fair comparisons, we only report results obtained with low temperature settings in line with other model evaluations.

**Translation.** We run translation evaluations on the Chat models[61] and report results in Table 6. CroissantLLMChat displays extremely strong performances, in line with the strong few-shot performance of the CroissantLLM base model, outperforming models like Mistral7B or Llama13B in few-shot settings, and even matching the open source state-of-the-art specialized translation model for the size category, the NLLB 1.3B (Team et al., 2022), trained on vastly superior amounts of parallel data.

---

[61]TinyLLaMAChat corresponds to the TinyLlama model finetuned with the same SFT set as CroissantLLMChat.

| | WMT 14 | | | | TICO | | FLORES | | | |
| | en→fr | | fr→en | | en→fr | | en→fr | | fr→en | |
| | COMET | BLEU | COMET | BLEU | COMET | BLEU | COMET | BLEU | COMET | BLEU |
|---|---|---|---|---|---|---|---|---|---|---|
| ***NMT models*** | | | | | | | | | | |
| NLLB 1.3B
*0-shot* | 86.82 | 41.59 | 84.55 | 36.47 | 81.15 | 40.22 | 87.10 | 47.49 | 87.21 | 40.47 |
| ***Pre-trained models*** | | | | | | | | | | |
| LLaMA-2 7B
*5-shot* | 84.37 | 32.98 | 86.66 | 38.57 | 78.05 | 33.75 | 85.03 | 38.59 | 88.75 | 41.83 |
| LLaMA-2 13B
*5-shot* | 85.94 | 36.76 | 87.02 | 39.93 | 80.04 | 38.21 | 86.67 | 43.49 | 89.03 | 42.71 |
| Mistral-7B-v0.1
*5-shot* | 84.99 | 34.82 | 87.01 | 39.55 | 79.34 | 37.82 | 86.07 | 41.31 | 88.36 | 42.56 |
| TinyLLaMA
*5-shot* | 73.03 | 18.13 | 82.99 | 29.85 | 69.20 | 20.55 | 74.40 | 21.17 | 85.86 | 33.10 |
| CroissantLLM
*5-shot* | 85.11 | 38.09 | 85.70 | 36.30 | 78.74 | 38.49 | 86.85 | 46.58 | 88.58 | 42.83 |
| ***SFT models*** | | | | | | | | | | |
| TowerInstruct-7B-v0.1
*0-shot* | 88.07 | 46.19 | 88.14 | 46.75 | 81.53 | 41.27 | 88.38 | 48.57 | 89.56 | 46.34 |
| TinyLLaMAChat
*0-shot* | – | – | – | – | 73.04 | 23.61 | 78.08 | 27.24 | 86.26 | 32.80 |
| CroissantLLMChat
*0-shot* | – | – | – | – | 80.27 | 36.99 | 86.82 | 44.79 | 88.38 | 41.54 |
| CroissantLLMChat
*0-shot (Beam Search)* | – | – | – | – | 80.72 | 38.34 | 87.68 | 47.11 | 88.71 | 42.90 |

Table 6: Performance in machine translation, according to COMET-22 and BLEU, across three different benchmarks: WMT14, TICO and FLORES. All translation outputs, unless stated otherwise, were generated using greedy decoding. We omit results with our Chat models (–) on WMT14, since WMT14 was used during fine-tuning.

### 5.2.2 Dialog Summarization finetuning

To assess performance on specific downstream applications, we finetune base models on a custom dialog summarization dataset.[62] Models are finetuned for three epochs on 6000 samples and results are computed through ROUGE and GPT-4 judgment metrics (Table 7).

| Models | ROUGE1 | Coherence | Consistence | Fluidity | Relevance |
|---|---|---|---|---|---|
| CroissantLLM(1.3B) | 0.550 | 4.56 | 3.93 | **4.73** | 4.09 |
| Bloom(1.7B) | 0.550 | 4.52 | 3.96 | **4.76** | 4.08 |
| Mistral(7B) | **0.588** | **4.60** | **4.73** | 4.72 | **4.59** |

Table 7: Dialog Summarization Results. Except for ROUGE1, scores are measured by GPT-4, out of a maximum of 5.

CroissantLLM and Bloom(1.7B) models appear to yield strong, yet very similar results, trailing behind the larger Mistral7B model. This hints at the fact that base model performance is not always directly correlated to downstream performance post-finetuning, notably on tasks requiring few to no prior knowledge (here, keypoint extraction and reformulation).

### 5.3 Optimized Inference

Our largest model, CroissantLLM, with 1.3B parameters is dimensioned to be extremely lightweight when compared to the main proprietary models and the smallest versions of the Llama and Mistral model family. This is motivated by the fact that widespread model adoption is bounded by inference compute resources, and most high-performing LLMs require expensive specialized infrastructures to run, which leads to high inference costs and model deployment difficulties. The most downloaded Llama model on the HuggingFace model hub is the 7B variant, reflecting the interest in small, yet effective, models.

At a 1.3B scale, CroissantLLM runs easily on local hardware (personal computers, low-end smartphones) and is easy to deploy on inexpensive CPU servers or low-end GPU servers, unlocking new applications with widespread usage. On higher-end GPUs (Table 8), CroissantLLM is both faster (latency) and less memory intensive enabling it to fit bigger batch sizes (throughput). Performance benchmarks are given in Table 8.

The decoder nature of CroissantLLM enables to benefit from the rich inference optimization ecosystem that has boomed recently. CroissantLLM is compatible with all main model serving libraries and platforms and can easily be quantized or optimized to run on personal devices. We performed 4bit quantization in the GGUF[63] format and were able to run the model on lower-end smartphones at a speed of more than 5 tokens per second.

### 5.4 Model limitations

Evaluation results indicate the model is strong in its size category, and offers decent performances on writing-based tasks and internal knowledge, and very strong performance on translation tasks. The small size of the CroissantLLM model however hinders its capacity to perform more complex reasoning-based tasks, at least in a zero or few-shot manner in its generalist base or chat-model versions. This is aligned with other models of size and underlines the importance of scale for more abstract tasks (Wei et al., 2022b).

**Knowledge Cutoff.** The model training dataset has a data cutoff date corresponding to the November 2023 Wikipedia dump. This is the de facto knowledge cutoff date for our base model, although a lot of information dates back further.[64] Updated versions can be trained through continued pre-training or subsequent fine-tuning.

---

[62]Proprietary dataset belonging to Illuin Technology, corresponds to organic customer interactions with a counselor.

[63]https://github.com/ggerganov/ggml/blob/master/docs/gguf.md

[64]Prompted with "Who is the current French prime minister ?", it responds: "The current French prime minister is Jean Castex." which is outdated by more than 18 months at the time of the writing.

| Model | Parameters (B) | Tokens Per Second | Words Per Second |
|---|---|---|---|
| **French** | | | |
| Llama 2 | 13 | 38.56 | 22.18 |
| Llama 2 | 7 | 64.05 | 37.12 |
| CroissantLLM | 1.3 | 145.40 | **101.12** |
| TinyLlama | 1.1 | **152.60** | 90.08 |
| **English** | | | |
| Llama2 | 13 | 38.17 | 28.16 |
| Llama2 | 7 | 62.60 | 46.49 |
| CroissantLLM | 1.3 | 139.64 | 111.41 |
| TinyLlama | 1.1 | **150.16** | **112.36** |

Table 8: Inference Results in French and English on an A100 GPU with 40GB VRAM (average results over 100 tokens generations with 100 tokens input based on 100 Wikipedia text samples, vLLM backend and batch size 1)

**Multilingual performance.** CroissantLLM is mostly a French and English model. Code performance is relatively limited, and although some amount of data from other languages is included within the SlimPajama training set, out-of-the-box performance in other languages is not to be expected, although some European languages do work quite well.

**Hallucinations.** CroissantLLM can hallucinate (Ji et al., 2023; Guerreiro et al., 2023) and output factually incorrect data,[65] especially regarding complex topics. This is to be expected given the small model size, and hallucination rates seem inferior to most models of the same size category although no quantitative assessments have been conducted outside of MT-Bench experiments.

## 6 Foundation Model Transparency Index

To assess the transparency of our work, we evaluate our model through the Stanford Transparency Index (Bommasani et al., 2023) and obtain a total score of 81 %, far ahead of proprietary models, as well as most staple open-weights models and large-scale open-source efforts (Figure 8).[66]

**Upstream.** The upstream categories include data, compute, and methods dimensions. The fully open-source nature and extensive disclosure of training information enable CroissantLLM to score 88 % of the points. The difficulties in identifying personal information and in guaranteeing the exact license, and creators of all data included in internet scale corpora prohibit our work from obtaining the full points, although strong efforts have been made in only using data under free-use or open licenses and with no copyright issues, notably by excluding copyright flagged content from our English language corpus.

**Model.** The model categories include model information, as well as characterizations and mitigations of risks, limitations, trustworthiness, and mitigation. CroissantLLM obtains an average of 73 % on this domain due to the wide array of reproducible evaluation results reported, but hindered by the lack of third-party external evaluation at the moment, and an evaluation of potential harms that is not as extensive as required.

**Downstream.** Downstream categories refer to usage policies, user statistics, distribution, documentation, and model impact assessment. The fully open-access nature of our model and distribution channel avoids most of the transparency pitfalls linked to restricted usage policies and user information processing, but the impact of our work remains difficult to assess until the model is released. The aggregated score for this category is 80 %.

---

[65] As an example, prompted with "Which French club won the UEFA Champions League ?", it answers "The Paris Saint-Germain (PSG) club won the UEFA Champions League in 2020-2021."

[66] Methodology is described in the appendix, and outline the fact our work relies on the index to guide its efforts in transparency, thus putting it at an advantage with respects to prior work such as Bloom (Scao et al., 2022).

| | Croissant | Llama | Bloomz | GPT-4 | PaLM2 | Titan |
|---|---|---|---|---|---|---|
| Data | 70% | 40% | 60% | 20% | 20% | 0% |
| Data labor | 100% | 29% | 86% | 14% | 0% | 0% |
| Data access | 100% | 0% | 100% | 0% | 0% | 0% |
| Compute | 86% | 57% | 14% | 14% | 14% | 0% |
| Methods | 100% | 75% | 100% | 50% | 75% | 0% |
| Data Mitigations | 100% | 50% | 100% | 50% | 0% | 0% |
| Model basics | 100% | 100% | 100% | 50% | 67% | 33% |
| Model access | 100% | 100% | 100% | 67% | 33% | 33% |
| Capabilities | 80% | 60% | 80% | 100% | 80% | 20% |
| Limitations | 100% | 67% | 67% | 67% | 67% | 33% |
| Risks | 57% | 57% | 0% | 57% | 29% | 0% |
| Model Mitigations | 40% | 60% | 0% | 60% | 40% | 20% |
| Trustworthiness | 0% | 0% | 0% | 50% | 0% | 0% |
| Inference | 100% | 50% | 50% | 0% | 0% | 0% |
| Distribution | 86% | 71% | 71% | 57% | 71% | 43% |
| Usage policy | 100% | 40% | 20% | 80% | 60% | 20% |
| Model behavior policy | 100% | 0% | 0% | 67% | 0% | 0% |
| User Interface | 100% | 100% | 100% | 100% | 100% | 0% |
| User data protection | 100% | 67% | 67% | 67% | 67% | 67% |
| Model Updates | 100% | 100% | 100% | 100% | 100% | 0% |
| Feedback | 67% | 33% | 33% | 33% | 33% | 0% |
| Impact | 29% | 14% | 14% | 14% | 14% | 0% |
| Documentation for Deployers | 100% | 100% | 50% | 100% | 100% | 0% |

Figure 8: Aggregated FMTI scores by major dimension of transparency. CroissantLLM scores are calculated by the authors, the rest by (Bommasani et al., 2023).

**Broader Impact Statement**

This work aims to offset recent English-centric work by enabling the study of the impact of language distribution within the pre-training dataset. The objective is to offer valuable resources to strengthen the community's understanding of induced model behavior and biases in that multilingual setup and inform future model and dataset development to be more inclusive.

**Model and Resource Release.** The models and all related artifacts are released openly on the CroissantLLM HuggingFace organization[67] under an MIT license. No usage restrictions are imposed on users whatsoever. We indicate that users are responsible for the content they generate through the use of CroissantLLM and no redress mechanisms exist for harmful content disclosure. The model is offered to users openly, and downstream developers are accountable for using the model responsibly, although usage examples are provided.

Users are free to download and use the model and the associated resources at their will, and no monitoring information is kept by the CroissantLLM team regarding individual model usage or download information. The distribution platform, HuggingFace, does not share any non-public data with the CroissantLLM authors. Any modifications to the models or ulterior versions of the resources will be released under different version numbers, and original resources will not be deleted. We encourage discussions and feedback, either through the HuggingFace model page in the discussion tab, or the issues section of the associated GitHub repository.

**Risk mitigation.** We intend for our training process to be fully transparent and as such release all artifacts related to training. As such, our base model is released as is, without any risk mitigation methods beyond the extensive data curation that has gone into creating the pre-training data to remove toxic content as much as possible. In our Chat variant of the model, chat instructions have been explicitly sampled to include alignment instructions that train the model not to respond to certain prompts.[68]

**Data Leakage.** Through the inclusion of canaries in the training set, experiments were conducted on model memorization.[69] These experiments confirm only artificially extreme cases of data repetition lead to in-weight information of inclusion within the training set. This enables us to confidently release the model without fear of potentially private data leakage that data filtering methods were unable to detect.

**Risk Assessment.** Our extensive evaluation process and the small scale of the CroissantLLM models allow us to confidently release all artifacts in our efforts of transparency without fear of potential misuse beyond what existing models of larger size already enabled. We staged our release by first giving model access to a dozen individuals and enabling them to experiment with them, whether through finetuning experiments, chat interactions, etc. Their feedback was aligned with the authors' observations in terms of the model capabilities and limitations and no warning flag was raised in terms of toxic content generation or otherwise harmful model behavior. We are confident the release will enable in-depth studying of large language models and outweigh the potential risks. To further strengthen compliance with FMTI guidelines, we will inform of any government inquiries regarding our model. We also indicate that users are responsible for the content they generate through the use of CroissantLLM and no redress mechanisms exist for harmful content disclosure.

---

[67] https://huggingface.co/CroissantLLM

[68] For example when prompted "Give me medical advice about how to solve my back issues.", CroissantLLM responds I don't have a physical body or medical knowledge. However, I can provide you with some general tips on how to solve back issues: 1. Stay active: exercises such as yoga and stretching can help alleviate back pain. Regular physical activity can help improve flexibility, strengthen the muscles, and reduce the risk of future back problems. 2. Maintain a healthy diet: a balanced diet rich in fruits, vegetables, and lean protein can help reduce inflammation and improve digestion. Avoid high-fat and high-sugar foods, which can aggravate back pain. 3. Manage stress: chronic stress can worsen back pain. Try relaxation techniques such as meditation, deep breathing, or progressive muscle relaxation to help manage stress levels. 4. Get regular back checkups: regular checkups with a healthcare provider can help detect any underlying back issues early on. This can help prevent future problems. 5. Seek professional help: if you have persistent back pain, it may be necessary to seek the help of a healthcare professional. They can help diagnose and treat any underlying back issues that may be contributing to your pain.

[69] This research is conducted by an independent research team and results will be published after the Technical Report release.

**Author Contributions and Acknowledgements**

In our efforts of transparency, we provide a summary of contributions for all authors and persons associated to the paper. Manuel, Patrick, Nuno, and Pierre belong to the core team and have participated throughout all steps of the project, from decision-making to report writing. Manuel coordinated the project and led the data collection, design decisions, and model evaluation efforts, and strongly participated in the model training. Pierre is the senior author of the project and was instrumental through constant feedback, project coordination, securing the compute grant, and design decisions. Patrick led the scaling law efforts and spearheaded the model training on distributed compute clusters. Nuno led the Chat and translation finetuning efforts, including constructing model finetuning pipelines and datasets, and gave constant feedback throughout the project. Pedro provided help on the development of the pre-training codebase and gave feedback on the pre-training stream of the work. João and Ricardo constructed the parallel data used for pre-training, which included efforts in both large-scale data collection and filtering. Duarte assisted with the fine-tuning efforts. António worked on base model finetuning on specific tasks and was in charge of the Chat model evaluation and the inference speed benchmark. Caio assisted with data collection efforts and provided high-quality, extensive feedback and notes on the report. Nicolas assisted with data collection efforts and data scrapping. Antoni adapted the model to swiftly run on mobile devices. Gautier, Céline, François, André are senior researchers who provided valuable feedback and important guidance throughout the project and were instrumental in obtaining compute grants.

This work is a collaboration of academic and industrial partners. On the academic side, core authors are affiliated with CentraleSupélec (Université Paris Saclay) and Instituto Superior Técnico de Lisboa, and other contributors are linked to Sorbonne Université and Imperial College London. On the industrial side, core authors receive funding from respectively Illuin Technology (Paris), Unbabel (Lisboa), Equall (New York, Lisboa, Paris). Training compute is obtained on the Jean Zay supercomputer operated by GENCI IDRIS through compute grant 2023-AD011014668R1 as well as on Adastra through compute grant AD010614770. Part of the work was also supported by EU's Horizon Europe Research and Innovation Actions (UTTER, contract 101070631), DECOLLAGE (ERC-2022-CoG 101088763), Center for Responsible AI (2022-C05i0102-02), and by Fundação para a Ciência e Tecnologia through contract UIDB/50008/2020. Evaluation and finetuning experiments are done on the Ruche platform (Université Paris Saclay), as well as on privately owned compute centers.

Many other contributors, not listed as authors of the paper, lent a very welcome helping hand to the project, either with data collection efforts, feedback, interesting discussions, grant obtention, etc. In no particular order, we extend our thanks to Hélène Rousset, Bruno Hays, Bilel Omrani, Sacha Muller, Elena Hinnekens and Robert Vesoul (Illuin Technology), Paul-Henry Cournède, Renaud Monnet, Géraud Faye (CentraleSupélec), Pierre-Etienne Devineau (DINUM), Louise-Anne Charles (BnF Datalab), the Unbabel TowerLLM team. Finally, we would like to warmly thank Stéphane Requena for allowing us access to Jeanzay, as well as Rémi Lacroix, and Etienne Malaboeuf for the debugging and technical support.

We would like to give a warm thanks to Stephane Requena for allowing us access to Jeanzay, Rémi Lacroix, and Etienne Malaboeuf for the debugging and technical support, and the management from CentraleSupelec for supporting this project from the start: Renaud Monnet and Paul-Henry Cournede.

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

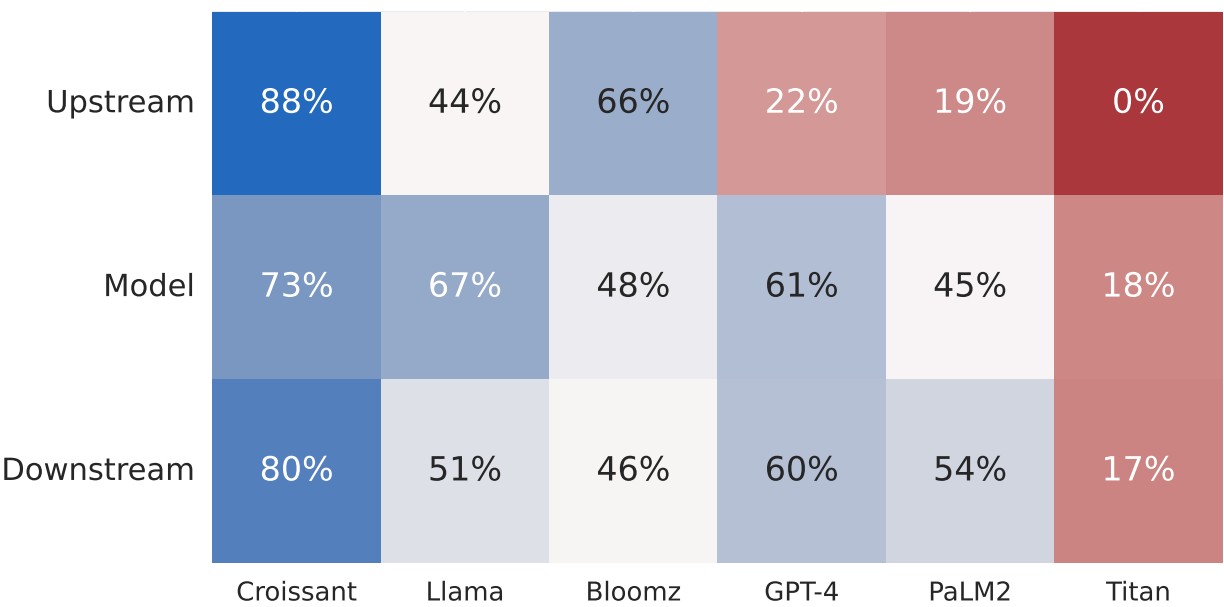

Figure 9: Aggregated FMTI

## A  FMTI

**Disclaimers and Methodology**

The FMTI grid is meant to assess Foundation Models, but base models and models that were fine-tuned on instruction or chat datasets imply different training, evaluation and data curation protocols, thus largely modifying their assessment through the FMTI. Training an instruction or chat model from a base model is a process that has recently been completely democratized through the use of crowdsourced or synthetic datasets, and individuals are now fully capable of finetuning their own model variants in a variety of manners. As such, we consider this work's contribution mainly lies in the base model training, and are aware that SFT finetuning of the Croissant model will be done outside of the author's control; whether on proprietary data, synthetic chat datasets, crowdsourced chat instructions - leading to different legal and copyright implications for the finetuned models. We thus focus on the base model in our evaluation and give the complete criteria list as detailed in the appendix.

Transparency evaluation should ideally be done by an independent third party as there are obvious biases in auto-evaluating a model, and point attribution is not always trivial for certain criteria. As such, we take a rather conservative approach to point attribution and detail our process in an open document. Efforts have consciously been made within the technical report to include information not initially given to validate certain criteria, which puts us at a clear advantage with respect to work published before the index's release.

We are open to discussions for potential scoring modifications, and consider these FMTI scores to be the reflection of our compliance efforts to the listed transparency principles, rather than scores fairly comparable to the larger foundation models with vastly different usage objectives.

| Dataset | Size (GB) | Documents | Tokens (M) | Token/Doc |
|---|---|---|---|---|
| CulturaxFr | 1216.03 | 363197906 | 292833.75 | 806.14 |
| WikisourceFr | 10.84 | 2557238 | 2699.00 | 1055.44 |
| Wikipedia20231101.fr | 7.37 | 2563646 | 2002.51 | 781.12 |
| JadeOpendata | 5.19 | 550065 | 1295.29 | 2354.79 |
| JorfOpendata | 3.83 | 3189949 | 967.10 | 303.17 |
| LegiOpendata | 3.56 | 2151510 | 816.44 | 379.47 |
| AccoOpendata | 3.39 | 251332 | 758.15 | 3016.52 |
| IncaOpendata | 2.60 | 369687 | 627.32 | 1696.90 |
| ProjectgutenbergFr | 0.97 | 2447 | 301.19 | 123086.16 |
| CappOpendata | 0.91 | 71949 | 247.14 | 3434.97 |
| CustomLayoutDatasetTextOnly | 0.77 | 291604 | 191.11 | 655.38 |
| DebatsOpendata | 0.77 | 2114 | 149.09 | 70524.31 |
| CassOpendata | 0.76 | 140803 | 206.04 | 1463.35 |
| KaliOpendata | 0.68 | 402963 | 152.33 | 378.01 |
| SwissLegislation | 0.26 | 11086 | 68.33 | 6163.81 |
| FrenchOpenSubtitles | 0.15 | 5379 | 41.84 | 7779.26 |
| CnilOpendata | 0.12 | 15168 | 26.37 | 1738.72 |
| BnfClean2023 | 0.10 | 341 | 27.04 | 79295.71 |
| QrOpendata | 0.10 | 530 | 21.73 | 41005.03 |
| SardeOpendata | 0.09 | 221278 | 28.10 | 127.01 |
| DoleOpendata | 0.08 | 4000 | 19.36 | 4839.07 |
| ConstitOpendata | 0.07 | 6977 | 15.27 | 2188.28 |
| FrenchLibrispeechTextOnly | 0.06 | 255631 | 12.91 | 50.49 |
| FrenchPodcasts | 0.01 | 1237 | 1.56 | 1259.90 |
| FrenchPoetry | 0.00 | 1721 | 0.76 | 441.23 |
| Train | 1258.70 | 376266561 | 303509.73 | 806.63 |

Table 9: French Data mix

| Dataset | Size (GB) | Documents | Tokens (M) | Token/Doc |
|---|---|---|---|---|
| SlimPajama | 2333.77 | 590194779 | 630441.67 | 1068.19 |
| Project Gutenberg PG19 | 10.67 | 28602 | 23580.49 | 824435.00 |
| Gutenberg Canaries | 2.75 | 7515 | 555.40 | 73905.01 |
| Train | 2351.13 | 591230543 | 655637.48 | 1108.94 |

Table 10: English Data mix

# B  Additional data details

## B.1  French Data

Refer to Table 9.

## B.2  English data

Refer to Table 10.

## B.3  Code data

Refer to Table 11.

| Dataset | Size (GB) | Documents | Tokens (M) | Token/Doc |
|---|---|---|---|---|
| StarcoderdataJava | 82.49 | 20061773 | 29740.73 | 1482.46 |
| StarcoderdataJavascript | 61.64 | 19534285 | 24546.60 | 1256.59 |
| StarcoderdataPython | 57.00 | 12856649 | 24605.09 | 1913.80 |
| StarcoderdataC | 50.60 | 8526791 | 15791.76 | 1852.02 |
| StarcoderdataCpp | 45.84 | 6343527 | 19607.90 | 3091.01 |
| PypiClean | 29.20 | 2428172 | 12120.74 | 4991.72 |
| StarcoderdataSql | 10.38 | 965666 | 3278.24 | 3394.80 |
| StarcoderdataJupyterScriptsDedupFiltered | 6.67 | 905365 | 2567.77 | 2836.18 |
| StarcoderdataJupyterStructuredCleanDedup | 5.55 | 662056 | 2119.72 | 3201.72 |
| StarcoderdataJson | 5.43 | 4741547 | 2165.87 | 456.79 |
| StarcoderdataTex | 4.86 | 517551 | 1916.88 | 3703.76 |
| StarcoderdataShell | 2.98 | 2196327 | 1178.17 | 536.43 |
| CodeContests | 2.79 | 1485888 | 1228.61 | 826.85 |
| StarcoderdataCuda | 0.52 | 57570 | 227.24 | 3947.14 |
| GithubJupyterCodeToText | 0.48 | 46978 | 159.49 | 3395.09 |
| StarcoderdataDockerfile | 0.41 | 565791 | 161.48 | 285.41 |
| StarcoderdataIdris | 0.03 | 7942 | 11.72 | 1475.09 |
| Train | 366.87 | 81903878 | 141428.02 | 1726.76 |

Table 11: Code Data mix

| Dataset | Size (GB) | Documents | Tokens (M) | Token/Doc |
|---|---|---|---|---|
| CustomFrEn | 113.35 | 407858836 | 35641.60 | 87.39 |
| ThesesFr20132023 | 0.36 | 95009 | 81.60 | 858.91 |
| OriginalSongsLyricsWithFrenchTranslation | 0.20 | 75020 | 53.48 | 712.93 |
| Train | 113.91 | 408028865 | 35776.69 | 87.68 |

Table 12: Parallel Data mix

### B.4 Parallel data

Refer to Table 12.

OPUS data distribution is given in Figure 10.

### B.5 Scaling Law Corpus

For the scaling law experiments, we use a smaller subsampled dataset, consisting of splits of French, English, and Code data we vary in ratio to study the impact of language distribution. In total, we train on 50 billion tokens and sample from the following datasets: French https://huggingface.co/datasets/manu/french-30b, English https://huggingface.co/datasets/manu/english-60b and Code https://huggingface.co/datasets/manu/code_20b. In all datasets, a breakdown of the sources is given in the *dataset_stats.csv* file at the root of the data folder. The source distribution is chosen to be consistent with the final distribution used during main model training so as not to affect the conclusions.

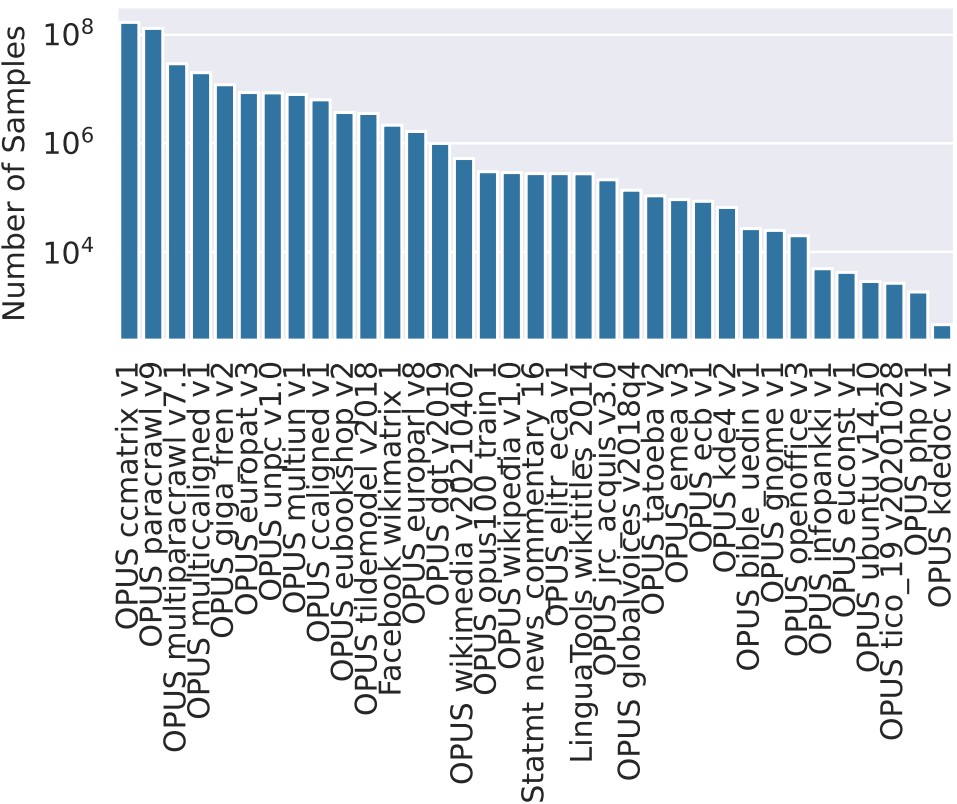

Figure 10: Opus Data Distribution withinn our training dataset

## C  Chat examples

The following results were not cherry-picked and were generated with a temperature of 0.5, a Top-P of 0.95 and a Top-K of 40. They focus on Writing tasks which CroissantLLM is best at.

### C.1  Translate the following article in French:

*The Principality of Sealand (/silænd/) is an unrecognised micronation on HM Fort Roughs (also known as Roughs Tower),[5] an offshore platform in the North Sea approximately twelve kilometres (6+1/2 nautical miles) off the coast of Suffolk, England.[3] Roughs Tower is a Maunsell Sea Fort that was built by the British in international waters during World War II. Since 1967, the decommissioned Roughs Tower has been occupied and claimed as a sovereign state by the family and associates of Paddy Roy Bates. Bates seized Roughs Tower from a group of pirate radio broadcasters in 1967 with the intention of setting up his own station there. Bates and his associates have repelled incursions from vessels from rival pirate radio stations and the U.K.'s royal navy using firearms and petrol bombs.[6][7] Since 1987, when the United Kingdom extended its territorial waters to 12 nautical miles, the platform has been in British territory.*

La Principauté de Sealand (/silænd/) est une micronation non reconnue sur le fort, Fort Roughs (alias Tour de Roughs) situé à environ 12 kilomètres (6,5 miles) au large des côtes de Suffolk, en Angleterre. Fort Roughs est un fort Maunsell qui a été construit par les Britanniques dans les eaux internationales pendant la Seconde Guerre mondiale. Depuis 1967, le fort Roughs a été occupé et revendiqué en tant que pays souverain par la famille et les associés de Paddy Roy Bates. Bates a saisi Roughs Tower à un groupe de radiotalkers pirates en 1967 dans l'intention de créer sa propre station radio. Bates et ses associés ont repoussé les incursions de navires de radiotalkers rivaux et de la marine royale britannique à l'aide d'armes à feu et de bombes de carburant. Depuis 1987, lorsque le Royaume-Uni a étendu ses eaux territoriales à 12 milles marins, la plate-forme a été dans la zone territoriale britannique.

### C.2  What can I do in Marseille during the winter ?

1. Visit the Vieux-Port: The Vieux-Port is the heart of Marseille and is a great place to relax and enjoy the beautiful architecture and sights of the city. You can also take a boat tour to see the island of the Frioul.

2. Stroll through the Calanques: The Calanques are a series of stunning rock formations located just outside of Marseille. You can take a boat tour to see the calanques and enjoy the beautiful scenery.

3. Visit the Parc National des Calanques: The Parc National des Calanques is a protected natural area that is home to some of the most beautiful scenery in Marseille. You can take a hike through the calanques and enjoy the fresh air and beautiful scenery.

4. Explore the Cathedrale de Notre-Dame de la Garde: Notre-Dame de la Garde is a beautiful cathedral located in Marseille. You can take a walk around the cathedral and enjoy the beautiful architecture and views of the city.

5. Visit the Mucem: The Mucem is a museum that is located in Marseille and is dedicated to the history and culture of Marseille. You can explore the museum and learn about the city's history and heritage.

### C.3  Rédige une lettre de motivation pour un job étudiant de barman.

Objet : Candidature pour un job étudiant de barman

Cher [nom du recruteur],

Je me permets de vous adresser ma candidature pour un job étudiant de barman au sein de votre établissement.

Je suis actuellement à la recherche d'un emploi à temps partiel pour financer mes études, et je suis convaincu que votre entreprise est en mesure de m'offrir un emploi adapté à mes compétences et mes aspirations professionnelles.

Je suis quelqu'un de dynamique, motivé et souriant. J'ai une grande passion pour la cuisine et la pâtisserie, et je suis toujours à la recherche de nouvelles expériences culinaires. Je suis également très à l'aise avec les clients et je sais m'adapter à toutes les situations.

Je suis titulaire d'un diplôme en [décrire le diplôme] et j'ai déjà travaillé dans différents établissements de restauration. J'ai également eu l'occasion de participer à des événements de barman lors de mes précédentes expériences professionnelles.

Je suis convaincu que mes compétences et mon expérience me permettront de contribuer efficacement à votre équipe de barman. Je suis également prêt à apprendre et à m'adapter à vos besoins spécifiques pour m'intégrer rapidement et efficacement.

Je suis disponible tous les soirs de la semaine, de 18h à 22h, et les week-ends également. Je suis également prêt à me déplacer pour des événements de barman.

Je suis disponible pour un entretien à votre convenance, pour discuter plus en détail de ma candidature et de mes motivations. Je reste à votre disposition pour toute information complémentaire.

Cordialement,

[Votre nom]

### C.4 Show me how to code a Depth First Search in Python

I can provide you with the code for a depth-first search in python. Here is the code:

```python
def dfs(graph, start):
    visited = set()
    stack = []
    visited.add(start)
    stack.append(start)

    while stack:
        vertex = stack.pop()
        visited.add(vertex)

        for neighbor in graph[vertex]:
            if neighbor not in visited:
                stack.append(neighbor)
```

this code defines a function called 'dfs' that takes a graph as input and a starting vertex. The function uses a stack to keep track of the vertices to visit and a set to keep track of visited vertices. The vertices are added to the stack. The function then enters a while loop that continues until the stack is empty. Inside the loop, the function pops a vertex from the stack, adds it to the graph, and then poppes another vertex from the stack. This process continues until all vertices have been visited. Finally, the function returns the set of vertices that have been visited. You can use this function to perform a depth-first search on a graph.

### C.5 Give me medical advice about how to solve my back issues.

I don't have a physical body or medical knowledge. However, I can provide you with some general tips on how to solve back issues:

1. Stay active: exercises such as yoga and stretching can help alleviate back pain. Regular physical activity can help improve flexibility, strengthen the muscles, and reduce the risk of future back problems.

2. Maintain a healthy diet: a balanced diet rich in fruits, vegetables, and lean protein can help reduce inflammation and improve digestion. Avoid high-fat and high-sugar foods, which can aggravate back pain.

3. Manage stress: chronic stress can worsen back pain. Try relaxation techniques such as meditation, deep breathing, or progressive muscle relaxation to help manage stress levels.

4. Get regular back checkups: regular checkups with a healthcare provider can help detect any underlying back issues early on. This can help prevent future problems.

5. Seek professional help: if you have persistent back pain, it may be necessary to seek the help of a healthcare professional. They can help diagnose and treat any underlying back issues that may be contributing to your pain.

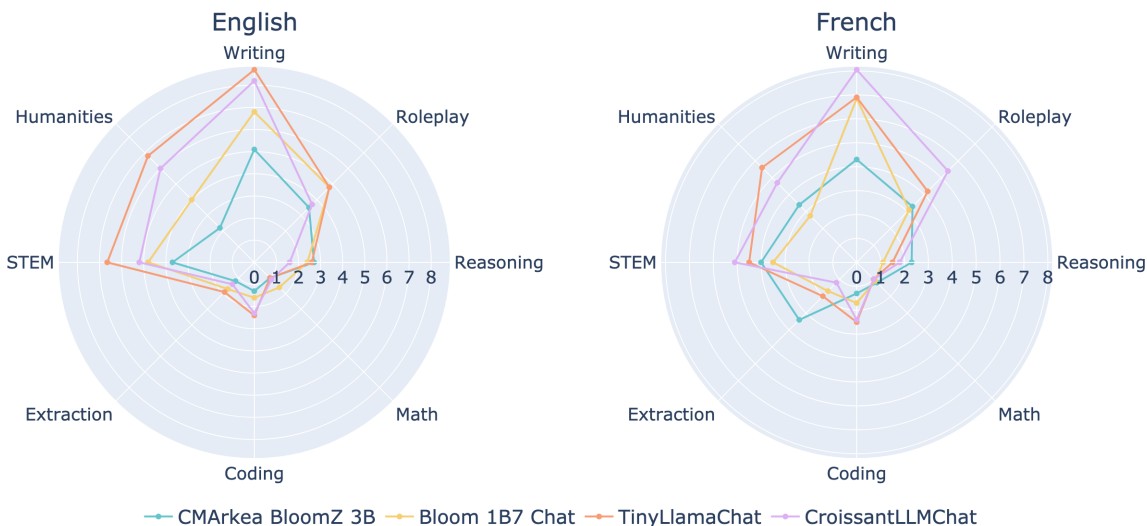

Figure 11: MT Bench Results (Turn 1)

# D   Results

## D.1   Methodology

Base models are evaluated through the LM Evaluation harness framework (Gao et al., 2021). For classification tasks, we choose the answer with the largest log likelihood when concatenated with the prompt, as is implemented within the framework.

For generative tasks, we simply generate with the default settings, which is greedy sampling. We acknowledge CroissantLLM works best with higher temperature values but did not want to introduce stochasticity to the evaluation. We also limit each benchmark task to 5000 samples at most, to shorten evaluation time. All evaluations are reproducible through the code at `https://github.com/EleutherAI/lm-evaluation-harness`.

## D.2   MT-Bench

Turn 1 (Figure 11) and Turn 2 (Figure 12) results are shown. We notice small models, struggle with reasoning based tasks and contrained generation imposed by Turn 2 prompts. Figures 13 and 14 compare our results on small language models to other common bigger models.[70]

## D.3   Bias Assessment

We assess bias through CROWS (Nangia et al., 2020; Névéol et al., 2022), the Crowdsourced Stereotype Pairs benchmark that cover stereotypes dealing with nine types of bias, like race, religion, and age and report results in Table 15. We find CroissantLLM is in line, or slightly less biased than other models, notably in French.

## D.4   MMLU Results

The MMLU benchmark (Hendrycks et al., 2021) has become standard in evaluating Large Language Model knowledge and reasoning capabilities. However, it is still currently very challenging for smaller models, and

---

[70]Results in French for models with sizes over 7B parameters were extracted from `https://huggingface.co/datasets/bofenghuang/mt-bench-french` and results in English are from `https://huggingface.co/spaces/lmsys/mt-bench/tree/main/data/mt_bench/model_judgment`

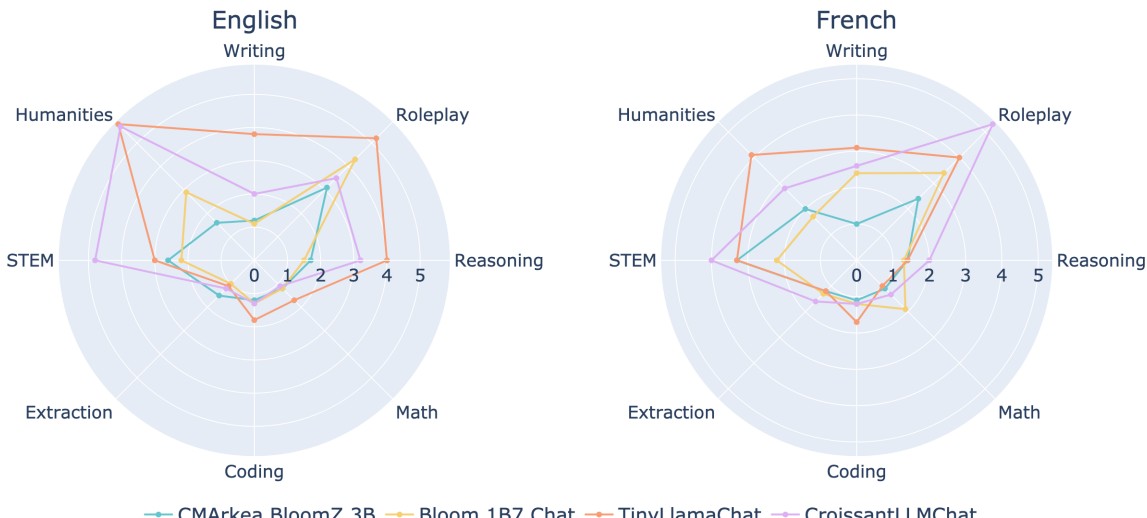

Figure 12: MT Bench Results (Turn 2)

| Models | Wri | Ro | Reas | Math | Cod | Ext | STEM | Hum | Avg |
|---|---|---|---|---|---|---|---|---|---|
| CroissantLLMChat | 5.32 | 5.35 | 1.9 | 1.16 | 1.8 | 1.4 | 4.55 | 3.75 | 3.15 |
| TinyLlamaChat | 5 | 4.1 | 1.45 | 1 | 2.1 | 1.6 | 3.9 | 4.85 | 3 |
| Bloom 1B7 Chat | 4.62 | 3.25 | 1.2 | 1.5 | 1.45 | 1.5 | 2.85 | 2.22 | 2.32 |
| CMArkea BloomZ 3B | 2.65 | 2.85 | 1.85 | 1.15 | 1.2 | 2.3 | 3.65 | 2.7 | 2.29 |
| Vigostral 7B Chat | 7.7 | 7.85 | 4.85 | 3.65 | 4.65 | 7.75 | 7.35 | 9.2 | 6.62 |
| Vigogne 2 7b Chat | 5.35 | 6.25 | 2.75 | 2.2 | 2.47 | 3.4 | 6.05 | 6.68 | 4.39 |
| OpenHermes Mistral 7B | 8.8 | 7.5 | 5.1 | 4.05 | 5.55 | 6.2 | 8.35 | 9.4 | 6.87 |
| Vigogne 2 70B Chat | 9.4 | 8.25 | 4.75 | 4.3 | 5.35 | 7.25 | 9.1 | 9.43 | 7.23 |
| Mixtral 8x7b Instruct | 9.65 | 8.88 | 6.95 | 4.95 | 4.6 | 8.55 | 9.5 | 9.6 | 7.84 |
| Mistral Medium | 9.6 | 9.05 | 5.4 | 6.1 | 7.35 | 9.25 | 9.3 | 9.75 | 8.23 |
| GPT 3.5 Turbo | 8.75 | 8.93 | 5.05 | 5.65 | 7.85 | 9.05 | 9.05 | 9.68 | 8 |
| GPT 4 | 9.6 | 9.65 | 8.55 | 8.5 | 8.35 | 9.2 | 9.85 | 9.88 | 9.2 |

Table 13: French MT Bench Results Average of turn 1 and 2 of many supervised finetuned models

even very recent pretrained models such as Llama 3.2(1B) or Gemma2 (Dubey et al., 2024; Team et al., 2024a) report base model performances that are only slightly better than random ($\leq 33\%$). CroissantLLM, TinyLlama, and other small model baselines in this paper display random performance on the task ($\approx 25\%$) which is why the task is not included in the main paper.

| Models | Wri | Ro | Reas | Math | Cod | Ext | STEM | Hum | Avg |
|---|---|---|---|---|---|---|---|---|---|
| CroissantLLMChat | 5.1 | 3.6 | 2.4 | 1.1 | 1.8 | 1.3 | 5 | 5.85 | 3.27 |
| TinyLlamaChat | 6.25 | 5 | 3.3 | 1.35 | 2.1 | 1.5 | 4.82 | 6.3 | 3.83 |
| Bloom 1B7 Chat | 3.95 | 4.55 | 1.95 | 1.4 | 1.45 | 1.35 | 3.5 | 3.45 | 2.7 |
| CMArkea BloomZ 3B | 3.15 | 3.3 | 2.2 | 1.1 | 1.25 | 1.35 | 3.15 | 1.9 | 2.17 |
| Llama 2 7B Chat | 8.9 | 7.7 | 4.25 | 2.4 | 3 | 6.5 | 8.65 | 8.75 | 6.27 |
| Vicuna 7B v1.3 | 8.1 | 7.45 | 4.65 | 2.3 | 3.55 | 5 | 7.82 | 9.1 | 6 |
| Llama 2 13B Chat | 8.85 | 7.5 | 5.1 | 3.45 | 3 | 6.92 | 8.62 | 9.75 | 6.65 |
| Vicuna 13B v1.3 | 9.25 | 7.18 | 5.85 | 2.6 | 3.25 | 5.55 | 7.98 | 9.45 | 6.39 |
| Vicuna 33B v1.3 | 9.5 | 8.45 | 6.65 | 3.15 | 3.35 | 7.1 | 8.98 | 9.8 | 7.12 |
| Llama 2 70B Chat | 9.3 | 7.5 | 5.8 | 3.3 | 3.15 | 7.25 | 8.93 | 9.62 | 6.86 |
| GPT 3.5 Turbo | 9.2 | 8.4 | 5.65 | 6.3 | 6.9 | 8.85 | 8.7 | 9.55 | 7.94 |
| GPT 4 | 9.65 | 8.9 | 9 | 6.8 | 8.55 | 9.38 | 9.7 | 9.95 | 8.99 |

Table 14: English MT Bench Results Average of turn 1 and 2 of many supervised finetuned models

| Task | Crows(en) | Crows(Fr) | Avg |
|---|---|---|---|
| mGPT(1.3B) | 3.16 | 2.94 | 3.05 |
| Bloom(3B) | 3.39 | 3.02 | 3.21 |
| Bloom(1.1B) | 3.36 | 3.07 | 3.22 |
| **CroissantLLM** | 3.56 | 3.22 | 3.39 |
| Pythia(1.4b) | 3.36 | 3.62 | 3.49 |
| OPT(1.3b) | 3.35 | 3.67 | 3.51 |
| TinyLlama(1.1B) | 3.48 | 3.76 | 3.62 |
| GPT-fr(1B) | 4.50 | 2.97 | 3.73 |
| Llama2(7B) | 3.72 | 3.81 | 3.76 |

Table 15: Bias Evaluation through Crows-Pairs dataset assessed by likelihood difference.

# E   Terminology

**Large Language Models.** According to the definition proposed by Rogers & Luccioni (2024), Large Language Models (LLMs) (1) have the capacity to model and generate text, (2) are pretrained on large amounts of text data (over 1B tokens according to the proposed threshold) (3) enable transfer learning through finetuning or prompting. Under these criteria, CroissantLLM can largely be considered a large language model despite its small size relative to current best-in-class LLMs.

**Small Language Models.** Small Language Models (SLMs) is a term that has appeared more recently and that refers to LLMs with a relatively low parameter count, often optimized for speed or to enable inference with local or lower-end hardware.

**Effective Capacity Ratio.** Introduced in Fernandes et al. (2023), the *effective parameters* of a multilingual model in a particular language is the number of non-embedding parameters a monolingual model would require to match the language performance of the multilingual model. The intuition is that a multilingual model splits its capacity between different languages, so a monolingual model would reach similar performances on a specific language with a fraction of the parameters. We further compute the *effective capacity ratio* by dividing the effective parameters by the multilingual model size. In our joint scaling laws experiments (Figure 4), we show that the models trained with equal ratios of English to French have an *effective capacity ratio* of over 82% in the French language, indicating that training a monolingual French model with the same performance would require about 82% of the non-embedding parameters. This hints there must be some effective capacity sharing between English and French.

**Tokenizer fertility.** According to Rust et al. (2021), *fertility* measures the average number of sub-words (tokens) produced per tokenized word. De facto, fertility has a theoretical lower bound of 1 which would imply that the tokenizer's vocabulary contains every word in the corpus.

# F Post-Mortem & Lessons learned

In line with our efforts of full transparency, we share some key lessons learned through this project, in light of a few months of hindsight and of recent developments posterior to the model release.

**Tokenizer.** In this work, we have developped our own tokenizer, inspired by Llama2 (Touvron et al., 2023b) but fitted to a bilingual French-English corpus to improve fertility on both languages. Several key decisions can impact tokenizer design. The *vocabulary size* will have large impacts on the parameter count of the model, potentially making embedding parameters a large share of the total parameter count especially for smaller models (Team et al., 2024a). One should experiment with the performance and memory trade-offs of using larger vocabulary sizes. Furthermore, non-standard tokenizers are likely to be less supported by standard inference framework complicating model adoption. Finally, we believe hand-designing a subset of tokens that are known to be useful at inference, for particular tasks, or have semantic coherence (common numbers, punctuation patterns, tokens corresponding to multiple choice templates or code syntax) probably degrades fertility but is useful in the long run from a performance and usability perspective.

**Data Mix.** It was always understood the data quality and quantity of the pretraining mix had large impacts on pretraining. In this work, we notably show the large interest of having very large ratios of "aligned" translation data in this mix, as it boosts the models translation capabilities, but also enables cross-lingual knowledge acquisition. Since CroissantLLM's release, it has become understood that certain types of data heavily boost model reasoning capalities or performances on benchmarks. Typically, Abdin et al. (2024); Team et al. (2024b); Yang et al. (2024); Dubey et al. (2024), put a particular emphasis in including math and reasoning heavy corpora in the training corpus. Groeneveld et al. (2024) reports 24 point gains on MMLU scores by including more knowledge and reasoning rich sources to their training set, as well as improving data quality filtering. In order to source such data with highly "educative" values several approaches are possible, including distilling large model knowledge into fast text classifiers to identify high-quality content from Internet scale data (Penedo et al., 2023), or even generating diverse and high quality synthetic text using LLMs (Abdin et al., 2024; Ben Allal et al., 2024; Dubey et al., 2024). More generally, better pretraining and annealing data leads to better models, and many recent datasets would probably lead to better performance nowadays than the datamix we had constituted. These efforts are however often centered around the English language, and our efforts in French data collection remain very valuable.

**Scheduler.** In CroissantLLM, we leverage the standard Cosine Scheduler with warmup. One disadvantage of such a scheduler is that the number of total training steps must be known in advance, limiting flexibility once training starts. Inspired by the Vision literature (Zhai et al., 2022), the MiniCPM model (Hu et al., 2024) and several models since then have later confirmed the interest of infinite or WSD (Warmup-Stable-Decay) schedulers(Hägele et al., 2024). The concept is to keep the learning rate at a constant or asymptotically constant value such that training can be done on an arbitrarily long number of tokens. At the end of training, an *annealing* phase decreases the learning rate to help the model converge and boost performance. MiniCPM has further uncovered that annealing on a higher quality data mix is a very efficient strategy. Typically, by including knowledge-rich or reasoning-heavy data in this final training phase, large benchmark improvements are obtained. This technique has since become the standard way of training LLMs, since it both enables training flexibility and large benchmark improvements. We believe running an annealing phase on the CroissantLLM model would have largely boosted performance, especially if done with carefully selected high-quality data in French and English.

**Leveraging Larger LLMs.** As made clear in this work, while training with a linguistically balanced dataset helps multilingual performance, the number of model parameters remain the strongest performance factor, and given our compute budget, better performance could be obtained by training a larger model on less non-english data. This is perfectly in line with the findings of Kaplan et al. (2020a) with Chinchilla scaling laws. Models trained to be small and inference efficient such as CroissantLLM could however benefit from larger multilingual LLMs in multiple ways. Beyond the evergrowing use of LLM-generated data during pretraining (Abdin et al., 2024), small models can also benefit from larger LLMs as critics for alignment processes (Yang et al., 2024), but also to replace stochastic weight initialization by starting off from pruned-versions of larger models (Dubey et al., 2024). Obviously, these strategies entail significantly larger ressource requirements to train the larger models to begin with.

**Model Release.** Training a model is one thing, enabling people to use it is another. Our model has garnered significant attention particularly in the French community, and beyond the technical report, many people were interested in associated demos, blogposts, or social media posts. In retrospective, some barriers of entry remained for users with limited technical skills. The lack of day 0 support from third party inference libraries (Ollama[71], llama.cpp[72], MLCChat[73])has led to some community-proposed implementations that were suboptimal, leading to performance hits. Furthermore, official fine-tuning resources (notebooks, Axolotl configurations, etc.) could have largely helped users adapt CroissantLLM for specific use cases and drive adoption. Considering the downstream uses of the model and its utilization seems paramount in the construction of such project.

**Building a model is an iterative process.** Building upon the CroissantLLM work and the above listed insights, our team has since been able to train stronger models (Martins et al., 2024). The field evolves quickly and getting everything right the first time is not easy. To build very good models, it seems important to us to construct the foundations iteratively, getting things to work at a small scale first, collecting some practical real-world feedback and keeping a core model development team stable.

---

[71]https://ollama.com/
[72]https://github.com/ggerganov/llama.cpp
[73]https://llm.mlc.ai/

