# OpenReview forum: "CroissantLLM: A Truly Bilingual French-English Language Model"
_TMLR — Accepted by TMLR_

### Review · Reviewer_RC4S · 2024-05-14

**Summary Of Contributions:**

The main contribution of the paper is an introduction of a new LLM featuring the equal emphasis of two languages, English and French, with data and a full recipe to reproduce the results. A new benchmark to assess the capability on French is also introduced. The data mixture is carefully designed to have good performance on both languages and the empirical findings and analyses are presented. The experimental results and findings are probably considered as not surprising and they are as expected -- the new LLM is better on the French benchmark while keeping good performance on English.

**Audience:**

Yes

**Broader Impact Concerns:**

No concerns.

**Claims And Evidence:**

Yes

**Requested Changes:**

There is no critical requested change for acceptance and these are all suggestions.

I would like to see the contributions mentioned in Introduction more clearly correspond to the concerns raised in the first part of introduction. I would like to see what the paper did to address the concerns. The contes are OK, but this is just a matter of structure.

I would like to see the exact definitions of used metrics in the paper at least as appendices, e.g. tokenizer fertility, capacity ratio, instead of giving brief explanations.

It would be good to give thoughts on the importance of the work for larger model sizes.

**Strengths And Weaknesses:**

The strengths are the contributions to the community by setting a great baseline and releasing all possible materials from the work. The introduction of the new benchmark would also be valuable to measure models' multilingual capabilities. In other words, the contributions are mainly made by providing a basis to perform further research with the materials released with the paper rather than novel scientific findings. There are some quantitive analyses of the model in the paper and those may be informative, but it seems to be that existing theories and knowledge can explain the results and there is no critical scientific finding.

The main weakness is the lack of scientific novelty. It is certainly a significant effort to collect new data and train a new large language model of a particular interest. However, it is a yet another effort to create a new model without novel findings.

Another weakness is that it is not clear if the discussions in the paper are only applicable to the class of models of the size or it is broadly applicable to extremely large models. The experimental results suggest that large models are better on French even with the distorted distributions. In that case, another approach would be to distill from a large model while keeping the desired multilingual performance as much as possible. There is no discussion on this.

---

> ### Author Response · Authors · 2024-10-18
>
> We first would like to thank the reviewer for their extensive read of our work and their valuable feedback.
>
> **Novelty**
>
> Beyond extensive efforts in introducing new relevant non-english benchmarks and large scale datasets which we feel are significant contributions, we agree that model checkpoints in themselves do not constitute an artifact that is particularly groundbreaking. The open-nature of the training recipe however constitutes a useful resource that has already enabled various posterior work, including work on model memorization (ICML 24) and various other pre-training projects. Recontextualizing the work, CroissantLLM is one of the first efforts to pretrain “small” models as an end goal, even though this has become more common since. Similarly, this projects predates other LLM pretraining efforts such as Olmo, LLM360,or SmolLLM, making the transparent nature of the work quite uncommon at the time within this training regime or in multilingual settings. Beyond a ressource release, we believe this work has uncovered or confirmed a few non-trivial findings.
> * Training on a more linguistically varied pre-training data distribution expands model internal knowledge, showcasing diversifying the pretraining data goes beyond language proficiency.
> * Reducing the share of English data does not significantly affect model performance on downstream tasks.
> * Training a model on such a large token to model parameter ratio does not lead to loss saturation, and models keep on improving. This result is contrary to Chinchilla scaling laws which indicate gains should be marginal in this regime. Such ratios have since become more common with the Llama3 release but were novel at the time.
> * Training with large amounts of translation data in the pretraining enables very strong translation performance, but also leads to a model that is able to store information in a common language subspace and leads to cross-lingual knowledge transfer. We show that on the translation task, generalist models trained as such can rival specialist translation models, as well as much larger models.
>
> **Other approaches**
>
> As the reviewer states, models with larger numbers of parameters will tend to perform better than smaller ones, even if trained on largely smaller multilingual data splits. This is aligned with Chinchilla Scaling Laws, and visible in our multilingual scaling laws ablations. Given our objectives of training an efficient, inference-optimal model, several follow-up approaches could be considered.
> The proposed distillation approach is partially what we do for finetuning. We collect synthetic  French and English samples generated by a much larger multilingual model and train through a causal language modeling objective on this data. We elected not to do this during pretraining to prevent legal issues that could arise by using model generated content. Additionally, since the training corpus of such models is not known, this would weaken our claims of transparency and potentially bias our experiments…
> A potential solution to derisk this process could be to use such data at the very end of pretraining in an “annealing” phase, as has been proposed by posterior work (MiniCPM) and is now common.  As seen in even more recent work, the best way to obtain a very strong model seems to be through larger model truncation & distillation (Llama 3.2 1b). Obviously, attempting doing this from scratch would require training larger models first and require significantly more compute resources.
> Other strategies for introducing multilingual capacities have been proposed in posterior work such as large scale continued pretraining (TowerLLM).
> As suggested by reviewer DVTq, we have added a section (Appendix Section F) highlighting the lessons learned from this project, including takeaways acquired in hindsight given the continued progress of the field after the release of our model. We use this section to include thoughts on leveraging larger models.
>
> **Requested Changes**
>
> * Definitions have been added in Appendix E and referenced in the main paper.
>
> * We add a clarifying sentence at the end of the introduction after the contributions to clarify how our work addresses the aforementioned challenges (in red).
>
> * As mentionned above, we discuss how leveraging larger models can be done in such a project given our findings and recent developments in Appendix section F.
>
> **Miscellaneous**
>
> We are interested in the reviewers opinion of whether the newly added Appendix Section F (lessons learned / hindsight a few months later) would be valuable to move in the main paper ?
>
> We thank the reviewer once again for their precious suggestions and comments and hope our modifications have strengthened our paper.

---

> > ### Comment · Reviewer_RC4S · 2024-12-13
> > **Thank you for the updates.**
> >
> > My concerns were addressed by the updates.
> >
> > I think that it is a good idea to incorporate the discussion in Appendix F in the main text.

---

### Review · Reviewer_DVTq · 2024-06-06

**Summary Of Contributions:**

This paper introduced CroissantLLM, a bilingual pre-trained LLM with bilingual proficiency in English and French. Despite its small size (1.3B), CroissantLLM was trained on a large corpus of bilingual English & French dataset (3T tokens), resulting in a small, efficient, yet high-performing model compared to what a training-compute-optimal model of a similar size would perform (Chinchilla scaling law; Hoffmann et al., 2022).

Unlike many closed-sourced LLMs (GPT-4, etc.), this paper open-sources many aspects of the model, from the training dataset, pre-trained checkpoints at various stages of training, and fine-tuned chat models, among others. The extent of the open-sourcing done in this paper (i.e. not just the pre-trained checkpoint, but also some of the training data, etc.) exceeds that of other open-sourced initiatives, such as LLaMa and Mistral 7B. Empirical results on multiple English and French benchmarks (including on machine translation benchmarks between the two languages) indicate that the model matches or outperforms other pre-trained models of similar sizes, although overall performance still generally lags behind larger models (e.g. Llama 2 7B & Mistral 7B), which are more costly to run.

**Audience:**

Yes

**Broader Impact Concerns:**

The broader impact concerns are already quite comprehensive. In my understanding, the authors already did their best in ensuring that the released training dataset complies with the relevant copyright regulations, and excluded copyright-restricted material from the model training and the released dataset.

However, I would like to ask a hypothetical (yet not entirely unrealistic) question: What happens if, later, it turns out that some of the released dataset contained copyright-restricted material? Would it be sufficient to simply remove that part from the publicly-released dataset? Or does this also mean that the model, which was trained on this copyright-restricted material, would need to be taken offline and retrained? If it is the latter, there might be a huge cost to retraining the model, etc. I would like to better understand how such situations, if they arose at a later time, would be handled. This hypothetical scenario attests to the importance of "unlearning" (e.g. Yuanshun Yao et al., 2023, the NeurIPS 2023 Machine Unlearning Challenge, etc.).

**Claims And Evidence:**

Yes

**Requested Changes:**

# Big Picture Changes (Essential)

1. Reassess whether or not the term "LLM" is appropriate for the model size that the paper is building, and if so, why.

2. Distill the key lessons learnt and recommendations from the authors' experience in building CroissantLLM. This will help others in the community who want to do similar things, for instance those who want to build LLMs for other languages. This could cover data collection advice, handling of "spike loss", overtraining factor, etc.

# Answering Questions / Suggestions (Not strictly essential but would strengthen the work)

1. On page 4, section 2.1, it seems that the authors collect data from Francophone countries / regions, for instance parts of Belgium, Switzerland, Senegal, Morocco, etc. How different are these news sources in terms of dialects, and how are these different dialects handled? Or are all the data sources mixed in without any special handling for different dialects?

2. On page 10, section 3.4, it seems that the authors did a careful handling of the bilingual data distribution, in order to "balance out" the English and French data. But each language is also comprised of multiple data sources, which can also differ wildly in terms of the size of each potential data source (e.g. French-English thesis abstracts likely contain much fewer tokens than French news sources). Is there any particular handling of the **domain imbalance** within a particular language?

3. On page 10, section 3.6, it seems that the authors observe loss spikes, which are quite common when training language models (more so for larger models). However, from my understanding, the authors simply ignored these loss spikes and let the model recover on its own. In my experience, these loss spikes can have a detrimental effect on the final loss. Have the authors tried simple things like reloading the checkpoint prior to a loss spike, and resuming training on a different batch of data? Or clipping the gradients, limiting the norms of the parameters, etc.? These techniques might mitigate the loss spikes, and ultimately yield a better final performance.

4. I don't see the English MMLU results on the tables in the main paper (please correct me if I'm wrong). Would it be possible to include them in order to facilitate a good comparison with prior work?

5. The French Trivia dataset (page 12, section 4.2.1) is described to use English questions. Why is this the case, and why not just frame the questions in French? Is it to test bilingual generalisation, or something else?

6. It would be good to put the model sizes of CroissantLLM on all the tables (e.g. Tables 3, 7, 8 etc.), because all the other models have the model sizes right next to them. This would facilitate an easier comparison.

# Typo

On page 7, "highly performing" should be "high-performing"?

**Strengths And Weaknesses:**

# Strengths

1. The model scores highly on the foundation model transparency index (FMTI), open-sourcing not just the final pre-trained checkpoint, but also a large part of the training data, checkpoint across different sizes of training, the overall codebases, and other relevant information. This transparency helps democratise the LLM scientific process, affording the community a chance to scrutinise and analyse the performance of the model across different checkpoints, and also replicate the authors' results and ultimately improve upon them. This point is becoming increasingly important, especially given the growth of closed-sourced LLMs, in addition to open-sourced LLMs that only shares the final parameters of the pre-trained model, without much detail as to what data the model was trained on, what the exact architecture looks like, etc.

2. The released model would likely be useful not just for scientists and engineers that would like to use the model, but also for end users who would like to use small yet efficient small models (in the order of ~1 billion parameters, for instance due to hardware constraints) that performs well despite its size, or for use cases that require good bilingual English / French LLM capability.

3. The bilingual angle (here French and English) is also interesting, as it helps combat the English-centricity of some prior open-sourced LLMs.

4. The small size of the model (only 1.3B parameter) is useful for broader adoption, because not many people (outside of academic and industry labs) have the hardware to run much larger models than this. The model was also trained with a high overtraining factor, which yields a much better performance than what a training-compute-optimal model of a similar size would achieve.

5. The training data collection process is described in detail, and includes creative uses of possible data sources. I particularly like the inclusion of bilingual thesis abstracts (page 6) that are written in both English and French, which leverages a useful domain (scientific literature).

6. There is a clear discussion about the environmental impact of the work, and how the model was trained on data centres that used low-carbon nuclear electricity (section 3.7, page 11).

7. From what I can tell, the authors took a conscious approach to potential test data contamination, for instance by excluding machine translation results with the chat model, which used WMT14 for fine-tuning and thus would suffer from test data contamination (Table 6, page 18).

# Weaknesses

1. I am not sure the model technically qualifies as an LLM, where the first "L" stands for "large", given that the model size is only 1.3B parameters. I do think that the model will be useful for end users, especially for those with hardware constraints (e.g. those with only 1 low-end or mid-end GPUs), in addition to scientists or engineers who conduct research on the scientific aspect of language models. My objection has more to do with the fact that the term "LLM" is a misnomer for models of this size.

2. The model focuses on bilingual capability on English and French. Both of these languages are high-resource languages, and are already fairly well-covered by current LLMs (including both closed-sourced ones and "open-sourced" ones like Llama or Mistral 7B that only open-source the final weights of the model). For this reason, French speakers likely already have a plethora of LLMs to choose from (especially for those that do not have hardware constraints that limit them to models of sizes ~1 billion parameters); it might be nice to include other, lower-resource languages that are not yet well-covered by current LLMs.

3. It would be nice to include more general lessons learnt / recommendations that are more generally applicable to those of us in the community who are building LLMs. For instance, what is the right "overtraining factor" to use? How does the model loss evolve if we train the model for, say, 10x / 20x / 50x / 100x what is training-compute-optimal (Hoffmann et al., 2022)? When does overtraining give only marginal gains, or when does performance start to saturate (if any)? Note that I am not asking the authors to redo the experiments; I am only asking them to outline and distill the key lessons learnt and recommendations for others who want to build similar LLMs, say for other languages.

---

> ### Author Response · Authors · 2024-10-18
>
> We thank the reviewer for their incredibly thorough review and relevant questions. We further address the few questions/suggestions.
>
> **Naming: “Large” Language Model**
>
> As the reviewer points out, whether a 1.3B parameter model qualifies as a “large” language model is not immediately obvious. We choose to abide by the definition proposed in the ICML 2024 paper by [Rogers, 2024] Position: Key Claims in LLM Research Have a Long Tail of Footnotes, in which the qualifying criteria is dependant on model capabilities and the pretraining data volume (over 1B training tokens) rather than the model size. Other criterias include the capacity to model and generate text, as well as enabling transfer learning through prompting or downstream finetuning. Our model trained on 3T tokens amply fits all of the criterions.
>
> More recently, and mostly after the public release of CroissantLLM, more and more models of reduced size but trained on trillion token corpora have started appearing due to their many benefits (Qwen2, Llama3.2, Gemma2, EuroLLM). At the time, it was not obvious these models had enough capacity to benefit from such large training tokens to parameter ratio. and the fact CroissantLLM was still significantly improving after 3T tokens was an interesting result of this work !  The term “Small Language Model” (SLM) has since become more common. We updated the manuscript to explicit these definitions in Appendix E.
>
>
> **Focus on lower ressource languages**
>
> Beyond a pure resource paper, this work aims to improve our understanding of model performance when trained on a language distribution largely different than the one common in most LLMs (in which English is predominant). By attempting to remove as much english-bias as possible by treating English and French similarly from a tokenization, data, training, and evaluation perspective, we are able to gain unbiased insights into the relationship between language distribution and language-specific performance (Section 3.3), but also an understanding of how culture-specific knowledge acquisition is aided by a more balanced training corpora (Section 5.1).
> For these experiments, it was essential to focus on languages with large high quality training data sizes in order to reduce potential biases associated to training on lower quality data from low resource languages. Furthermore, while other work has focused on highly multilingual models (Bloom, Aya), focusing on 2 languages only enables our model to perform well on both, rather than trading off performance to cover a wider range of languages.
>
> **Key Lessons**
>
> We thank the reviewer for his suggestion. This long review process has actually been quite beneficial to understand the impact of certain decisions and we have updated the paper to include some lessons & takeaways we find impactful in hindsight ! As this section is a bit anachronic with the rest we have added it to the Appendix in section F. If the reviewer feels this is valuable, we can also move this to the main paper in the final version.
>
>
> **What happens if, later, it turns out that some of the released dataset contained copyright-restricted material?**
>
> This situation is probably one of the reasons many large models are released openly but without their training set (Llama, Mistral). In our case, we have put in place 3 mechanisms to deal with this. First, a particular emphasis has been put in excluding all potentially copyrightable content from our training set. Secondly, data deletion requests are easy to take into account through our hugging face data repository, and we are happy to report that 10 months later, although our models have collectively been downloaded over 150K times, no such deletion request has ever been filed. Finally, and perhaps most interestingly, through the inclusion of data “canaries”; ie. sequences of texts we have generated ourselves and added within the training set, we were able to run controlled experiments into model memorization, and to gain insights into the types of sequences and the number of repetitions necessary for the model to significantly remember certain documents. While this is the basis of another research paper, since then accepted to ICML 2024 and that we will link to upon acceptance (we didn’t before for anonymity reasons), this gave us concrete evidence that the inclusion of specific data samples could not particularly impact the final model independently. Beyond copyrighting issues, this enables us to confidently release our model knowing no PII data would be regurgitated by the model. We will mention this work in the camera ready version but refrain from doing this now to preserve anonymity.
>
>
> Once again, we thank the reviewer for their review, have fixed the suggested typo, and hope our response and paper modifications shines a light on the few questions they had !
>
> As the review was very detailed, we respond to the additional suggestions in a following comment.

---

> > ### Author Response · Authors · 2024-10-18
> >
> > **Suggestions:**
> >
> > 1. We have included data from various french-speaking countries, but regional dialects that are too linguistically different from the french language have not been included. Contrary to certain languages such as Arabic or Portuguese in which the language largely differs depending on the country, the written French language in Canada, Belgium, Switzerland, Senegal, Morocco, Lebanon, France is largely similar with only a minor share of regionally introduced syntax and expressions. As such, we did not consider any special handling of these sources. We modified the paper to explicit this (Section 2.1).
> >
> > 2. While we have controlled to have similar amounts of monolingual English and French tokens, it is true the source distribution within each language can slightly differ. In both languages, the data is predominantly filtered internet data, to which is added literature, encyclopedic content such as wikipedia, and other high-quality data sources. Since data sources are not always available in similar quantities in both languages, we have decided to favor training with the highest quality data we could source across both languages. We also make sure to include in the pretraining set very large proportions of “aligned” data (translations) which by nature have exactly the same semantic distribution across both languages.This is an attempt to help cross-lingual generalization from an early stage, thus reducing biases introduced by different source distributions.
> >
> > 3. Previous literature rarely mentions handling of such issues. Some models have reloaded training by skipping batches (OPT, PALM), others have seemed to ignore it if loss recovered (Llama1). In our case, we used gradient clipping, weight decay and tuned optimizer values (OPT, Llama) to minimize the occurrences of such loss spikes…  During training, as spikes were few and far between, and loss seemed to recover quite rapidly every time and without visible consequences, we opted to let the model train with no intervention. We highlight that since our checkpoints and dataloaders are openly available, this enables future experimentations around these loss spikes. We explicit this in the paper (Section 3.6).
> >
> > 4. At the time of writing, MMLU was considered a benchmark ill-suited for such small models, both because of the formatting scheme that was poorly understood by models of the size, and because it is challenging, requiring very specific knowledge and reasoning capabilities. Our model (and all baselines of the size in the paper) display random performance on the task (25%). This has slightly evolved, with more recent small models, typically obtained through model truncation and distillation on synthetic data, having been able to yield very slightly better-than-random performance on MMLU (Llama3.2 1B claims 33.2%). We now mention this in the appendix in section D.
> >
> > 5. The reviewer touches on a key aspect of the French trivia task ! The questions pertain to french-related culture but are asked in English for two reasons:
> > * to make sure english models are not penalized. We want to assess the internal knowledge of similarly sized models such as TinyLlama or Pythia trained predominantly in english but prefer not to have their bad grasp of the french language bias results.
> > * Inversely, we want to make sure this knowledge is not just regurgitated as seen in the training data by CroissantLLM. Our hypothesis is that training on more french helps knowledge acquisition, but that the model is able to leverage the acquired knowledge when responding to and generating text in english showcasing knowledge is shared in a common latent space.
> >
> > 6. We have made the requested changes, thanks for the suggestions !

---

> > > ### Comment · Reviewer_DVTq · 2024-11-15
> > > **Thank You**
> > >
> > > Thank you for the thorough authors' response, and for addressing my feedback and suggestions. I have read both Appendices E & F, and found them to be good additions to the paper. I can confirm that my concerns have been addressed.

---

### Review · Reviewer_mYt3 · 2024-10-10

**Summary Of Contributions:**

- This paper is a detailed technical report about how to train and evaluate a English-French bilingual LLM.
- This paper open-sources its training data, final models and benchmark (especially a manual verified benchmark on French) for research in this area.
- This paper releases CroissantLLM, which leads performance on french language at the 1B scale in the experiments.

**Audience:**

Yes

**Broader Impact Concerns:**

The broader impact concerns of this work have been fully addressed in this paper, including model release, risk mitigation, data leakage and risk assessment, no issues are left.

**Claims And Evidence:**

Yes

**Requested Changes:**

As mentioned in weaknesses,

- Add more experiments about the behaviour of model under different ratios of French data (values, common sense, taboos, etc.). It is also good to build some related benchmarks to evaluate the "English-bias" of bilingual / multilingual models.
- Authors can add more baselines of 1B-scale open-source models, such as Qwen-2.5 and Gemma2 series.

**Strengths And Weaknesses:**

Strengths:
- This paper conducts an insightful discussion on the language ratio selection when training bilingual LLM, and take ablation studies to find the optimal ratio between English and French.
- Transparent and open source. Data, intermediate checkpoints and final models are all open. It is valuable for community to research the training dynamic of a bilingual LLM (English is not the only dominate language).
- FrenchBench, a high-quality human verified benchmark for french language, benifit the development of the French LLM.

Weaknesses:
- As a research paper, I think it is need more content to discuss the “Bias towards English” problem. It comes as no surprise that increasing the portion and amount of french pretraining data will lead to the performance on Frecnh-orient benchmark. But after increasing the proportion and quantity of French data, whether the model begins to be less "English-biased"—such as whether its responses align more closely with the values of the French-speaking community, better match the habits of French users, or reflect the living environment of French speakers—becomes a more worthy topic of exploration. These aspects can better argue the significant importance of building a non-English-dominant model (bilingual model).
- This paper lack some comparisons between some popular 1B-scale open-source models, such as Qwen-2.5 and Gemma2 series. These models also perform impressively in French.

---

> ### Author Response · Authors · 2024-10-18
>
> We thank the reviewer for their detailed read of our work and their review that underlines our contributions both in terms of openly released ressources (benchmarks, models, data, code) but also through the analysis and ablations we led to determine the impacts of language distributions.
> We further respond to the two weaknesses.
>
> **Language-dependent bias**
>
> In our work, we detail how pretraining with largely more equal ratios of english to french data affects the final model. We show several non-trivial key findings:
>
>
> * *Multilingual Scaling Laws*: Pretraining on 50% of French data largely improves benchmark performance on french benchmarks. However, this does not come at the expense of English benchmark performance. Through our scaling laws ablations, we are able to calculate relationships between the language distribution in the pretraining set and language specific performance.
> * *Knowledge acquisition*: Compared to the similar TinyLlama model, we show pretraining with large amounts of french data has a large impact on the cultural knowledge included in the model weight. Typically, french-culture related information is better assimilated, even after finetuning on similar data. This implies that beyond pure “language” performance, having more representative pretraining data matters to enhance varied model knowledge.
>
> The reviewer is further interested in exploring biases in model values. In this work, we have partially explored this topic through the CrowS-Pairs benchmark, and its French counterpart French Crows-Pairs (*Appendix D.3*). Crows-Pairs assesses the degree to which U.S. stereotypical biases are present in language models. We find CroissantLLM is in line, or slightly less biased than other similarly sized models, notably in French but draw no significant conclusions on the matter.
>
> We agree that beyond stereotypes, it would be very interesting to construct a full benchmark that extensively evaluates differences in philosophical beliefs, taboos, habits or values. However, such things differ even within a language community, and doing this correctly would require extensive crowd-sourcing, ressources and lies at the intersection of social science. Although we believe such work is a bit beyond the scope of our manuscript, the fact both the CroissantLLM models and the entire training datasets are openly available enable such work, and we are aware some research teams are currently using our model to conduct such experiments.
>
>
> **Baselines**
>
> The CroissantLLM model has been publicly released in January and the manuscript submitted to TMLR in March. At the time of writing, we had made sure to include all of the best available baselines of the size (and even bigger sizes such as Llama2-7B and Mistral 7B models). The models proposed by the reviewer Gemma2 and Qwen2.5 are undoubtedly strong baselines but have been respectively released mid-september, and late july. In fact, both the Qwen2, and Qwen1.5 series have also been released after the paper submission (and Qwen1 did not have similarly sized checkpoints). It is possible to add results for these models in the manuscript, but we are unsure of whether it is logical to do so… These models are not representative of the landscape at the time of the submission, and a better understanding of model pretraining / benchmark boosting (in parts due to the openly released CroissantLLM manuscript) will largely favor more recent models.
> As suggested by other reviewers, we have added a section with key takeaways and lessons learned in hindsight, highlighting evolutions in the field since the release of CroissantLLM (*Appendix Section F*). We believe this may be a way to add value to this 10 month old paper without introducing chronological inconsistencies or unfair baselines. We would be very interested in having the reviewers or the ACs opinion on if this is a satisfying way to handle this !
>
>
> We thank the reviewer again for their precious comments.

---

### Decision · Action_Editor_29NJ · 2025-01-15

**Recommendation:** Accept as is

**Comment:**

The paper has received positive feedback from all the reviewers. The reviewers appreciate the paper's contributions to the research community, particularly its open-source nature and focus on bilingual proficiency. The reviewers also highlight the paper's detailed technical report on training and evaluating the bilingual model, as well as its insightful discussion on language ratio selection and the creation of a high-quality benchmark for the French language. However, some reviewers suggest that the paper could benefit from more content discussing the "Bias towards English" problem and comparisons with other popular 1B-scale open-source models.

Overall, I recommend accepting the paper for publication due to its valuable contributions and practical applications.

**Audience:**

Yes

**Claims And Evidence:**

Yes